# Distribution-Free Fair Federated Learning with Small Samples

Qichuan Yin

University of Chicago

Zexian Wang

University of Michigan, Ann Arbor

Junzhou Huang

University of Texas at Arlington

Huaxiu Yao

UNC-Chapel Hill

Linjun Zhang *

Rutgers University

**Abstract**

As federated learning gains increasing importance in real-world applications due to its capacity for decentralized data training, addressing fairness concerns across demographic groups becomes critically important. However, most existing machine learning algorithms for ensuring fairness are designed for centralized data environments and generally require large-sample and distributional assumptions, underscoring the urgent need for fairness techniques adapted for decentralized and heterogeneous systems with small-sample and distribution-free guarantees. To address this issue, this paper introduces FedFaiREE, a post-processing algorithm developed specifically for distribution-free fair learning in decentralized settings with small samples. Our approach accounts for unique challenges in decentralized environments, such as client heterogeneity, communication costs, and small sample sizes. We provide rigorous theoretical guarantees for both fairness and accuracy, and our experimental results further provide robust empirical validation for our proposed method.

**Keywords:** Federated learning; Algorithmic Fairness; Distribution-free

**Mathematics Subject Classification (2020):** 62H30, 62R07

## 1 Introduction

Federated learning (FL) enables collaborative model training across multiple clients without requiring data centralization (McMahan et al., 2017). This paradigm has become increasingly important in applications involving sensitive data, such as healthcare (Joshi et al., 2022; Antunes et al., 2022) and mobile systems (Li et al., 2020; Yang et al., 2021). As FL systems are deployed in high-stakes settings, ensuring *algorithmic fairness* across demographic groups has emerged as a critical concern.

Despite extensive progress in fairness-aware machine learning, most existing methods are designed for centralized settings and rely on large-sample or distributional assumptions. Directly applying these approaches in federated environments is nontrivial and often leads to

---

*Corresponding author: linjun.zhang@rutgers.edu

degraded performance or excessive communication overhead. Moreover, the decentralized nature of FL introduces additional challenges, including client heterogeneity, limited local sample sizes, and restricted data sharing, all of which complicate the enforcement of fairness constraints.

Recent works have attempted to address algorithmic fairness in the FL setting, including FairFed (Ezzeldin et al., 2023), FedFB (Zeng et al., 2021), FCFL (Cui et al., 2021), and AgnosticFair (Du et al., 2021). These methods typically enforce fairness by modifying local training objectives or adjusting aggregation weights. However, they suffer from two fundamental limitations. First, fairness is primarily enforced at the local level, while achieving *global* fairness is inherently challenging: fairness at the client level does not necessarily translate to fairness at the population level (Hamman and Dutta, 2023). Second, most existing approaches rely on asymptotic guarantees or fail to provide fairness guar-

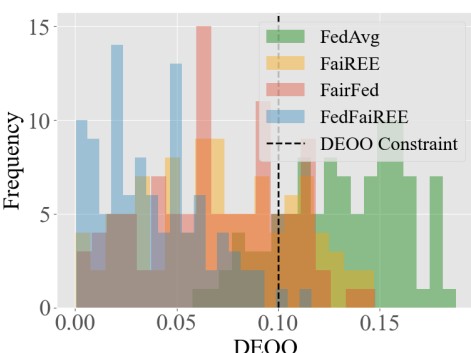

Figure 1: The distribution of $|DEOO|$ (defined in equation 2) for different methods on the Adult dataset (Dua et al., 2017). See Section 5 for details.

antees in a *distribution-free* manner where no distributional assumptions are imposed, limiting their applicability in realistic scenarios with small and heterogeneous datasets.

FaiREE (Li et al., 2022) is, to the best of our knowledge, the first method that provides group fairness guarantees that are both distribution-free and small-sample that ensures fairness with arbitrarily finite samples. However, it is restricted to centralized i.i.d. data and does not address the distinctive challenges of FL, such as decentralized data, communication constraints, and client heterogeneity. In heterogeneous FL settings, even if all training data are centralized, FaiREE can still suffer from bias due to cross-client distributional differences. These challenges are particularly pronounced in applications such as healthcare, where privacy regulations prevent data sharing and each institution only has access to a small, siloed dataset. This gap motivates the need for a method that achieves small-sample, distribution-free fairness guarantees while explicitly accounting for decentralization and heterogeneity.

To address these challenges, we propose FedFaiREE, a general post-processing framework for achieving *small-sample* and *distribution-free* fairness in federated learning. Our key insight is that controlling group fairness can be reduced to aligning order statistics of score distributions across groups, even under heterogeneous and decentralized data. This perspective enables us to transform fairness constraints into a rank-based selection problem, leading to both theoretical guarantees and practical efficiency. Our framework applies to a broad class of group fairness notions, including Equality of Opportunity (Hardt et al., 2016), Equalized Odds (Hardt et al., 2016), Demographic Parity (Agarwal et al., 2018), Predictive Equality (Hardt et al., 2016), and Overall Accuracy Equality (Zafar et al., 2017). The method is designed for realistic decentralized settings with heterogeneous client distributions, limited sample sizes, and communication constraints.

The key idea of FedFaiREE is to leverage distributed order statistics to construct a set of candidate classifiers that satisfy fairness constraints with high probability, and then select the

most accurate classifier within this feasible set. This formulation converts global fairness control into a rank-based problem across clients, enabling principled handling of heterogeneity and approximate local rank estimation under communication constraints. Importantly, this perspective decouples fairness enforcement from model training, leading to a flexible and theoretically grounded solution.

Our contributions are threefold:

1. We introduce FedFaiREE, a unified post-processing framework that takes any black-box classifiers as input and enforces group fairness constraints in federated learning under finite samples, client heterogeneity, and multiple protected groups. To the best of our knowledge, this is the first framework that provides distribution-free, small-sample fairness guarantees in the federated setting.

2. We establish rigorous guarantees showing that FedFaiREE satisfies fairness constraints with high probability in a distribution-free manner with arbitrarily finite sample, and achieves near-optimal accuracy under mild conditions on the input scorer.

3. Empirical validation. Through extensive experiments (Figure 1), we demonstrate that existing federated fairness methods often fail to control fairness under small-sample or heterogeneous settings, while FedFaiREE consistently satisfies the prescribed fairness constraints with competitive accuracy.

## 1.1 Additional Related Work

Fairness in federated learning has been studied from two complementary perspectives: fairness across clients and fairness across demographic groups. The former focuses on equitable performance or contribution across clients (Li et al., 2021; Lyu et al., 2020; Yu et al., 2020; Huang et al., 2020), while the latter aims to ensure equitable treatment across sensitive attributes such as race or gender, commonly referred to as *group fairness* (Dwork et al., 2012).

**Existing Group Fairness Techniques.** Existing approaches to group fairness can be approximately divided into three categories (Caton and Haas, 2020): pre-processing methods that directly perform debiasing on input data (Zemel et al., 2013; Johndrow and Lum, 2019); in-processing methods that incorporate fairness metrics into model training as part of the objective function (Goh et al., 2016; Cho et al., 2020); post-processing methods that adjust model outputs to enhance fairness (Li et al., 2022; Zeng et al., 2022; Fish et al., 2016). Our method falls into the post-processing category. Among these works, FaiREE (Li et al., 2022) is the most closely related, as it provides finite-sample, distribution-free fairness guarantees in the centralized setting. However, extending such guarantees to federated learning is nontrivial. In federated settings, optimizing local fairness objectives does not generally ensure global fairness(Hamman and Dutta, 2023). Instead, fairness must be enforced at the level of the global population, while the data remain distributed across clients with heterogeneous local distributions, limited local sample sizes, and communication constraints. To address this, we develop a distributed order statistics formulation that links global fairness control to client-level rank information, and further incorporate communication-efficient local rank estimation into the analysis. This yields finite-sample, distribution-free fairness guarantees in decentralized settings where FaiREE is not

directly applicable. In addition, our proposed method allows client correlation, while FaiREE requires independence among training samples. Our method is also applicable to a broader range of scenarios than FaiREE, including settings with label shift. See a more detailed discussion in Section D of the Appendix.

**Group Fairness Approaches in Federated Learning.** In recent years, there has been a growing amount of work focusing on group fairness in the context of Federated Learning (Ezzeldin et al., 2023; Cui et al., 2021; Zeng et al., 2021; Du et al., 2021; Rodríguez-Gálvez et al., 2021; Chu et al., 2021; Liang et al., 2020; Hu et al., 2022; Papadaki et al., 2022). Most of these studies aim to either introduce fairness principles into the local updates, adapt conventional fairness methods, or perform reweighting during aggregation, or a combination of these strategies. Specifically, Du et al. (2021) proposed AgnosticFair, a framework that utilizes kernel reweighing functions to adjust items in local objective functions, including both loss terms and fairness constraints. Zeng et al. (2021) introduced FedFB, a method that adapts Fair Batch, a centralized technique designed to improve fairness among groups by reweighting loss terms for different subgroups, for the FL setting. Ezzeldin et al. (2023) proposed FairFed, an approach that adjusts aggregate weights by considering the disparities between local fairness metrics and the global fairness metric in each training round.

## 2 Preliminaries

In this paper, we address the problem of predicting a binary label, denoted by $Y$, using a set of features. The features are divided into two categories: $X$ and $A$. Here, $X \in \mathcal{X}$ represents non-sensitive features, while $A \in \mathcal{A} = \{0, 1, \cdots, A_0\}$ corresponds to sensitive features. A data point includes $(x, y, a)$, which corresponds to $(X, Y, A)$. For simplicity, we first introduce the concept of *Score-based classifier* (Chen et al., 2018; Zafar et al., 2019).

**Definition 2.1.** (Score-based classifier) A score-based classifier is an indication function $\hat{Y} = \phi(x, a) = \mathbb{1}\{f(x, a) > c\}$ for a measurable score function $f : \mathcal{X} \times \mathcal{A} \to [0, 1]$ and a constant threshold $c > 0$.

To assess the fairness of the classifier, several group fairness notions have been proposed in the literature. In the following, for illustration, we introduce two commonly used notions, Equality of Opportunity (EOO) and Equalized Odds (EO). Both notions compare the probability of receiving a positive prediction across protected groups, but they condition on different subsets of the population. Equality of Opportunity focuses only on individuals whose true label is positive, while Equalized Odds additionally considers individuals whose true label is negative.

**Definition 2.2.** (Equality of Opportunity (Hardt et al., 2016)) A classifier satisfies Equality of Opportunity if it satisfies the same true positive rate among protected groups: $\mathbb{P}_{X|A=a,Y=1}(\hat{Y} = 1) = \mathbb{P}_{X|A=0,Y=1}(\hat{Y} = 1)$, where $a \in \{1, \cdots, A_0\}$.

Intuitively, Equality of Opportunity requires that, among individuals who truly belong to the positive class, membership in a protected group should not substantially affect the chance of receiving a positive prediction. This is fairness-relevant because, in many applications, a positive prediction corresponds to a favorable outcome or treatment, such as being classified as

eligible, qualified, or low-risk. For example, in the income prediction task, $Y = 1$ indicates that an individual has a high income. Equality of Opportunity requires that, among individuals who are truly high-income, the probability of being predicted as high-income should be comparable across protected groups.

**Definition 2.3.** (Equalized Odds (Hardt et al., 2016)) A classifier satisfies Equalized Odds if it satisfies the following equality: $\mathbb{P}_{X|A=1,Y=1}(\widehat{Y} = 1) = \mathbb{P}_{X|A=0,Y=1}(\widehat{Y} = 1)$ and $\mathbb{P}_{X|A=1,Y=0}(\widehat{Y} = 1) = \mathbb{P}_{X|A=0,Y=0}(\widehat{Y} = 1)$.

Compared with Equality of Opportunity, Equalized Odds is a stronger requirement because it controls disparities on both sides of the label distribution. In the Adult example, it requires not only that truly high-income individuals from different protected groups have comparable chances of being predicted as high-income, but also that truly non-high-income individuals from different protected groups have comparable chances of being incorrectly predicted as high-income.

In practice, exact equality is often unattainable. Therefore, a tolerance parameter, denoted as $\alpha$, is commonly introduced in Equality of Opportunity, as discussed in prior works (Zeng et al., 2022; Li et al., 2022). To be more specific, given a classifier $\phi$, the $\alpha$ difference tolerance in Equality of Opportunity within a binary group label can be defined as:

$$|\mathbb{P}_{X|A=1,Y=1}(\widehat{Y} = 1) - \mathbb{P}_{X|A=0,Y=1}(\widehat{Y} = 1)| \leq \alpha. \tag{1}$$

To be concise, in later sections, we use $DEOO$ to represent the left side of the inequality, i.e.,

$$DEOO = \mathbb{P}_{X|A=1,Y=1}(\widehat{Y} = 1) - \mathbb{P}_{X|A=0,Y=1}(\widehat{Y} = 1). \tag{2}$$

Similarly, the difference with respect to equalized odds can be defined as a two-dimensional vector

$$DEO = (\mathbb{P}_{X|A=1,Y=1}(\widehat{Y} = 1) - \mathbb{P}_{X|A=0,Y=1}(\widehat{Y} = 1), \mathbb{P}_{X|A=1,Y=0}(\widehat{Y} = 1) - \mathbb{P}_{X|A=0,Y=0}(\widehat{Y} = 1)). \tag{3}$$

We will extend our framework to multi-group settings in Section 6.2 later.

**Additional Notation.** To further simplify the formula in the article, we provide notations as follows: $p_a$ signifies $P(A = a)$. $p_{Y,a}$ represents $P(Y = 1 \mid A = a)$, and $q_{Y,a}$ is defined as $1 - p_{Y,a}$. $D$ and $D_i$ represent the datasets for all clients and client $i$, respectively, where $i$ belongs to the set $\{1, 2, \ldots, S\}$. $n$ denotes the size of dataset $D$. $T$ represents the ordered scores of elements in dataset $D$. $D_i^{y,a}$ is used to denote the subset of dataset $D_i$ where $Y = y$ and $A = a$. Similar notations apply to $T^{y,a}$ and $n^{y,a}$.

# 3 Fair Federated Learning Approach

In this section, we introduce FedFaiREE, a **Fed**erated Learning, **Fai**r, distribution-f**REE** algorithm. FedFaiREE is designed to ensure fairness under three key challenges: finite samples, distribution-free settings, and heterogeneous clients. Our key insight is that controlling group fairness can be reduced to aligning the *order statistics* (i.e., quantiles or ranks) of score distributions across sensitive groups. This perspective allows us to reformulate fairness constraints

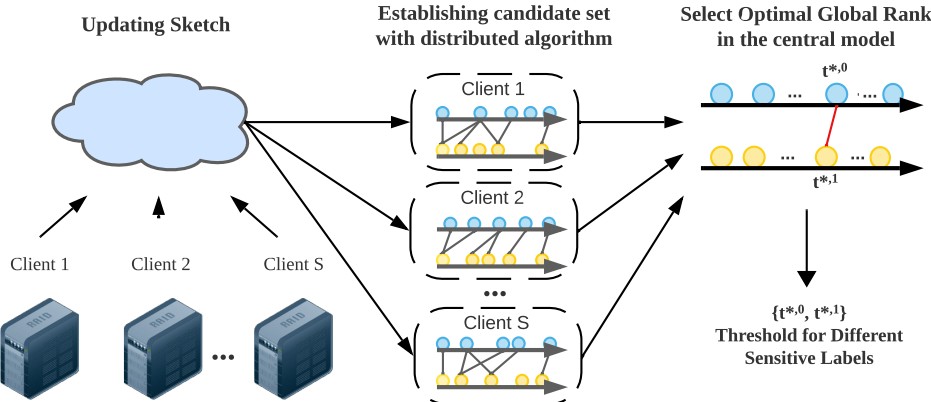

Figure 2: **Overview of FedFaiREE.** With S clients and a pre-trained model in consideration, each circle in the image symbolizes a datapoint score in the training set. The color of the circles represents different sensitive labels, while the gray edges depict local ranks of threshold pairs (each global classifier's threshold pair corresponds to S local ranks). Notably, the red edge signifies the chosen global classifier with thresholds $t^{*,0}, t^{*,1}$ for sensitive labels $A = 0$ and $A = 1$, respectively.

as a rank-based selection problem, which remains tractable even under heterogeneity and small samples.

To incorporate heterogeneity among clients, we adopt the following assumption.

**Assumption 3.1.** The training data points within the client $i$ are drawn independently and identically (i.i.d) from distribution $P_i$, while the test data points are sampled from a global distribution that represents a mixture of $P_1, \cdots, P_S$ with weight $\{\pi_i\}_{i \in [S]} \in \Delta_S$. Specifically, we assume that

$$\left(X_k^i, Y_k^i\right) \sim P_i, \quad \left(X^{\text{test}}, Y^{\text{test}}\right) \sim P^{mix} = \sum_{i=1}^{S} \pi_i P_i.$$

This implies that data points in client $i$ are exchangeable. Specifically, we want to note that we do not make any assumptions among $P_1, \ldots, P_S$ here.

## 3.1 Problem formulation

Consider a scenario with $S$ clients, each with a local dataset $D_i = \cup_{y \in \mathcal{Y}, a \in \mathcal{A}} D_i^{y,a}$ and a pre-trained score-based classifier $\phi_0(x, a) = \mathbb{1}\{f(x, a) > c\}$. Here, $D_i^{y,a}$ denotes samples with label $Y = y$ and sensitive attribute $A = a$. Our goal is to construct a fair classifier of the form

$$\phi(x, a) = \mathbb{1}\{f(x, a) > \lambda_a\},$$

where $\lambda_a$ is a group-specific threshold chosen to satisfy a fairness constraint such as $|DEOO| < \alpha$.

Piror work (Corbett-Davies et al., 2017; Menon and Williamson, 2018; Zeng et al., 2022) has shown that the classifier with optimal misclassification performance while adhering to specific fairness constraints requires group-wise thresholdings to the unconstrained Bayes-optimal classifier, we consider group-wise scores $t_{i,j}^{y,a} = f(x_{i,j}^{y,a}, a)$ and denote the sorted scores on client $i$ by $T_i^{y,a} = \{t_{i,1}^{y,a}, t_{i,2}^{y,a}, \cdots, t_{i,n_i^{y,a}}^{y,a}\}$. Let $T^{y,a}$ denote the global sorted scores aggregated across all clients. Instead of directly searching over thresholds, FedFaiREE takes a different perspective: we operate on the *ranks* of scores. On client $i$, this naturally leads us to the idea of

transforming the problem of selecting optimal thresholds $\lambda_a$ into determining the optimal "local ranks" $k_i^{1,a}$ of the score. However, as we concern about global fairness and misclassification error, we opt to seek the global rank $k^{1,a}$ (i.e., the rank in the sorted score set $T^{1,a}$), and $\phi(x,a) = \mathbb{1}\{f(x,a) > t_{(k^{1,a})}^{1,a}\}$. By mapping this to its corresponding "local ranks" $k_i^{1,a}$, we can leverage the properties of order statistics to ensure fairness under client heterogeneity. We will delve into the details of our approach and observations in the next subsection.

To this end, we present an overview of our algorithm in Figure 2, consisting of two main parts: 1). establishing a candidate set with a distributed algorithm that satisfies the fairness constraint with high probability, and 2). selecting the optimal rank pair to minimize the estimated misclassification error. In the next subsection, we first discuss a simple case: ensuring equality of opportunity under a binary-group and binary-label scenario, i.e., $\mathcal{Y} = \mathcal{A} = \{0,1\}$. As FedFaiREE is a general framework adaptable to various fairness notions and can accommodate even more diverse situations, we are going to study how to apply the method to equalized odds in Section 3.4, and extend to other fairness notions in Section B in the appendix. The extension to the multi-group fairness scenario will be further studied in Section 6.2.

**Scope of the distribution-free guarantee.** Throughout the paper, we use "distribution-free" to denote that a property holds with no distributional assumptions. Conditional on the pre-trained score function $f$ used for post-processing, the fairness guarantee does not assume a distribution family for the client-specific conditional score distributions $f(X,A) \mid Y = y, A = a$, nor does it rely on asymptotic approximations. Some parts of the theoretical presentation invoke continuity of the induced score distributions to simplify the order-statistic argument. However, this continuity condition is mainly imposed for expositional simplicity and is not essential, as discussed in detail in Appendix E.

## 3.2 Candidate set construction with distributed quantile algorithm

We now describe how to construct a set of candidate rank pairs that satisfy the fairness notion equality of opportunity (EOO). Since EOO depends on differences in group-wise true positive rates, and these rates correspond to tail probabilities of score distributions, they can be controlled by the relative positions (ranks) of thresholds within each group. Therefore, fairness can be enforced by selecting rank pairs whose induced quantiles are sufficiently aligned across groups. To formalize this, we leverage the theory of order statistics.

Specifically, we consider score sets that $k^{1,a}$ represents the rank in the sorted $T^{1,a}$. To account for heterogeneity among clients, we further introduce the notation $k_i^{1,a}$ to denote the corresponding rank of $t_{k^{1,a}}^{1,a}$ within the sorted set $T_i^{1,a}$, where $i \in [S]$ and $k_i^{1,a}$ satisfies $t_{i,(k_i^{1,a})}^{1,a} \leq t_{(k^{1,a})}^{1,a} < t_{i,(k_i^{1,a}+1)}^{1,a}$. For simplicity, we further define $\boldsymbol{k}^{1,a} = (k_1^{1,a}, \cdots, k_S^{1,a})$, and $Q(\alpha,\beta)$ represents independent variable following a Beta$(\alpha,\beta)$ distribution. We present the following observation regarding fairness control.

**Proposition 3.2.** *Under Assumption 3.1, for $a \in \{0,1\}$, consider $k^{1,a} \in \{1,\ldots,n^{1,a}\}$, the corresponding $k_i^{1,a}$ for $i \in [S]$ and the score-based classifier $\phi(x,a) = \mathbb{1}\{f(x,a) > t_{(k^{1,a})}^{1,a}\}$. Define*

$$h_{y,a}(\boldsymbol{u},\boldsymbol{v}) = \mathbb{P}\Big( \sum_{i=1}^S \pi_i^{y,a} Q\left(u_i, n_i^{y,a} + 1 - u_i\right) - \sum_{i=1}^S \pi_i^{y,1-a} Q\left(v_i, n_i^{y,1-a} + 1 - v_i\right) \geq \alpha \Big). \qquad (4)$$

*Then we have:*

$$\mathbb{P}(|DEOO(\phi)| > \alpha) \leq h_{1,0}(\boldsymbol{k}^{1,0} + \boldsymbol{1}, \boldsymbol{k}^{1,1}) + h_{1,1}(\boldsymbol{k}^{1,1} + \boldsymbol{1}, \boldsymbol{k}^{1,0}), \tag{5}$$

*where $\pi_i^{1,a} = \mathbb{P}(x \text{ from client } i \mid x \text{ with } Y = 1, A = a)$.*

Intuitively, Proposition 3.2 explains how we can assess fairness using ranks, instead of directly estimating the unknown population DEOO. For a threshold classifier, the rank of the threshold among the positive samples provides information about the corresponding true positive rate. In the federated setting, each client may have a different score distribution, so the same global threshold can correspond to different local ranks across clients. Proposition 3.2 combines these local ranks with the client weights and provides an upper bound on the probability that the DEOO constraint is violated.

This proposition enables us to select classifiers that satisfy fairness constraints with arbitrary finite samples and no distributional assumption. Moreover, $Q(\alpha, \beta)$ can be efficiently estimated by Monte Carlo simulations in applications. Specifically, we approximated $Q(\alpha, \beta)$ by conducting random sampling 1000 times in our experiment, yielding a highly satisfactory approximation.

Due to the need of computing local ranks to make use of Proposition 3.2, it is crucial to consider the tradeoff between accuracy and communication cost in real applications. We can adopt distributed quantile algorithms to reduce communication costs while controlling errors in calculating local ranks. Therefore, we present an alternative formulation of Proposition 3.2 to allow errors in the local rank calculation. To begin with, we introduce the concept of approximate quantiles and ranks (Luo et al., 2016; Lu et al., 2023).

**Definition 3.3.** *($\varepsilon$-approximate $\beta$-quantile and rank of a given set) For an error $\varepsilon \in (0, 1)$, the $\varepsilon$-approximate $\beta$-quantile of a given set is any element with rank between $(\beta - \varepsilon)N$ and $(\beta + \varepsilon)N$, where $N$ is the total number of elements in set. Further, the $\varepsilon$-approximate rank of an element in a given set is any rank between $(\beta - \varepsilon)N$ and $(\beta + \varepsilon)N$ where $\beta N$ represents the real rank.*

In other words, an $\varepsilon$-approximate rank or quantile may differ from its exact counterpart by at most an $\varepsilon$-fraction of the sample size, which provides a natural way to characterize the error introduced by communication-efficient distributed rank estimation. Under Definition 3.3, if the rank estimation method produces $\varepsilon$-approximate ranks, it is possible to correspondingly modify Proposition 3.2.

**Proposition 3.4.** *Under Assumption 3.1, for $a \in \{0, 1\}$, consider $k^{1,a} \in \{1, \ldots, n^{1,a}\}$, the corresponding $\hat{k}_i^{1,a}$ for $i \in [S]$ which are $\varepsilon$-approximate ranks and the score-based classifier $\phi(x, a) = \mathbb{1}\{f(x, a) > t_{(k^{1,a})}^{1,a}\}$. Define*

$$h_{y,a}(\boldsymbol{u}, \boldsymbol{v}) = \mathbb{P}\Big(\sum_{i=1}^{S} \pi_i^{y,a} Q\left(u_i, n_i^{y,a} + 1 - u_i\right) - \sum_{i=1}^{S} \pi_i^{y,1-a} Q\left(v_i, n_i^{y,1-a} + 1 - v_i\right) \geq \alpha\Big). \tag{6}$$

*Then we have:*

$$\mathbb{P}(|DEOO(\phi)| > \alpha) \leq h_{1,0}(\boldsymbol{M}^{1,0}, \boldsymbol{m}^{1,1}) + h_{1,1}(\boldsymbol{M}^{1,1}, \boldsymbol{m}^{1,0}), \tag{7}$$

where $\pi_i^{1,a}$ is defined in Proposition 3.2, $\boldsymbol{M}^{1,a} = (M_1^{1,a}, \cdots, M_S^{1,a})$, $\boldsymbol{m}^{1,a} = (m_1^{1,a}, \cdots, m_S^{1,a})$, $M_i^{1,a} = \min\left(\lceil \hat{k}_i^{1,a} + \varepsilon n_i^{1,a} \rceil, n_i^{1,a} + 1\right)$, $m_i^{1,a} = \max\left(\lceil \hat{k}_i^{1,a} - \varepsilon n_i^{1,a} \rceil, 0\right)$. Especially, $Q(0, \beta) = 0$ and $Q(\alpha, 0) = 1$ for $\alpha, \beta \neq 0$.

Proposition 3.4 shows that the fairness guarantee remains valid even when the exact local ranks are replaced by $\varepsilon$-approximate ones. The effect of rank approximation is captured through the inflated upper and lower rank bounds $\boldsymbol{M}^{1,a}$ and $\boldsymbol{m}^{1,a}$, which enlarge the uncertainty region in a conservative manner; as $\varepsilon$ decreases, this bound becomes tighter and recovers Proposition 3.2 in the exact-rank case.

In practical distributed settings, calculating the exact local rank in Proposition 3.4 is generally hard due to communication constraints. By adopting approximate $\varepsilon$ and related parameters in a distributed quantile algorithm, we strike a balance between accuracy and communication cost, enabling the effective implementation of our algorithm in distributed environments.

In our experiments, we implemented the Q-digest (Shrivastava et al., 2004), a tree-based sketching distributed quantile algorithm commonly used for efficiently approximating quantiles and ranks computation with rigorous theory controlling the error. Due to the inherent characteristics of the Q-digest algorithm, it only yields approximate quantiles and ranks that tend to be greater than their true values. However, considering the adaptability of other distributed quantile algorithms and aiming to reduce the absolute value of $\varepsilon$, we take into account both upward and downward estimation deviations as described in Definition 3.3.

By Proposition 3.4, we construct the candidate set $K$ as

$$K = \{(k^{1,0}, k^{1,1}) | L(\boldsymbol{k}^{1,0}, \boldsymbol{k}^{1,1}) < 1 - \beta\}, \tag{8}$$

where $\boldsymbol{k}^{1,a} = (\hat{k}_1^{1,a}, \cdots, \hat{k}_S^{1,a})$ are estimated corresponding "local ranks" of $k^{1,a}$, and $L(\boldsymbol{k}^{1,0}, \boldsymbol{k}^{1,1}) = h_{1,0}(\boldsymbol{M}^{1,0}, \boldsymbol{m}^{1,1}) + h_{1,1}(\boldsymbol{M}^{1,1}, \boldsymbol{m}^{1,0})$ corresponding to the right-hand side of the inequality equation 7.

## 3.3  Selection for the optimal threshold

In this subsection, we elaborate on our method for selecting the optimal threshold. For a given pair $(k^{1,0}, k^{1,1})$ from the candidate set, we exploit the properties of order statistics to compute the estimated misclassification error and then select the pair minimizing the estimated error.

To facilitate this, we need to compute the approximate ranks of $t_{(k^{1,0})}^{1,0}$ and $t_{(k^{1,1})}^{1,1}$ in the sorted sets $T_i^{0,0}$ and $T_i^{0,1}$, where $i \in [S]$, respectively. Specifically, we determine $k_i^{0,a}$ such that $t_{i,(k_i^{0,a})}^{0,a} \leq t_{(k^{1,a})}^{1,a} < t_{i,(k_i^{0,a}+1)}^{0,a}$ for $a \in \{0, 1\}$. To simplify, in the following sections, we assume the corresponding $\hat{k}_i^{1,a}$ for $i \in [S]$ are $\varepsilon$-approximate ranks and the estimated quantiles presented by distributed quantile algorithm are $\varepsilon$-approximate quantiles. Then, we commence by presenting our observation on the estimation of misclassification error through the following proposition.

**Proposition 3.5.** *Under Assumption 3.1, the misclassification error can be estimated through*

---

**Algorithm 1** FedFaiREE for EOO

---

**Input:** Calibration Dataset $D_i = D_i^{0,0} \cup D_i^{0,1} \cup D_i^{1,0} \cup D_i^{1,1}$; pre-trained classifier $\phi_0$ with function f; fairness constraint parameter $\alpha$ ; Confidence level parameter $\beta$; Weights of different clients $\pi$

**Output:** classifier $\hat{\phi}(x,a) = \mathbb{1}\{f(x,a) > t_{(k^{1,a})}^{1,a}\}$

1: **Client Side:**
2: **for** i=1,2,..,$S$ **do**
3:     Score on train data points in $D_i$ and get $T_i^{y,a} = \{t_{i,1}^{y,a}, t_{i,2}^{y,a}, \cdots, t_{i,n_i^{y,a}}^{y,a}\}$
4:     Sort $T_i^{y,a}$ and calculate q-digest of $T_i^{y,a}$ on client $i$
5:     Update digest to server
6: **end for**
7: **Server Side:**
8: Construct $K$ by $K = \{(k^{1,0}, k^{1,1}) | L(\boldsymbol{k}^{1,0}, \boldsymbol{k}^{1,1}) < 1-\beta\}$ {Establishing a set that satisfies fairness constraints and confidence requirements using order statistics. The search for $(k^{1,0}, k^{1,1})$ can be simplified using technique in Appendix C.1.}
9: Select optimal $(k_0, k_1)$ by minimizing equation 9 using estimated values $\hat{p}_a^i$, $\hat{p}_{Y,a}^i$ and $\hat{q}_{Y,a}^i$

---

$\hat{\mathbb{P}}\left(\hat{\phi}(x,a) \neq Y\right)$, which equals to

$$\sum_{i=1}^{S} \pi_i \Big[ \frac{\hat{k}_i^{1,0} + 0.5}{n_i^{1,0} + 1} p_0^i p_{Y,0}^i + \frac{\hat{k}_i^{1,1} + 0.5}{n_i^{1,1} + 1} p_1^i p_{Y,1}^i + \frac{n_i^{0,0} + 0.5 - \hat{k}_i^{0,0}}{n_i^{0,0} + 1} p_0^i q_{Y,0}^i + \frac{n_i^{0,1} + 0.5 - \hat{k}_i^{0,1}}{n_i^{0,1} + 1} p_1^i q_{Y,1}^i \Big]. \tag{9}$$

*Further, the discrepancy between empirical error and true error is upper bounded:*

$$\left| \mathbb{P}\left(\hat{\phi}(x,a) \neq Y\right) - \hat{\mathbb{P}}\left(\hat{\phi}(x,a) \neq Y\right) \right| \leq \theta, \tag{10}$$

*where $\theta = \sum_{i=1}^{S} \pi_i [e_i^{0,0} p_0^i q_{Y,0}^i + e_i^{0,1} p_0^i p_{Y,0}^i + e_i^{1,0} p_1^i q_{Y,1}^i + e_i^{1,1} p_1^i p_{Y,1}^i], e_i^{y,a} = \frac{2\lfloor \varepsilon n_i^{y,a}\rfloor + 1}{2(n_i^{y,a}+1)}.*

Proposition 3.5 provides a method for estimating the overall misclassification error using data from the training set with equation 9. However, we may not have exact knowledge of the probabilities $p_a^i$ and $p_{Y,a}^i$. In such cases, we can use the estimated values $\hat{p}_a^i = \frac{n_i^{0,a} + n_i^{1,a}}{n_i^{0,0} + n_i^{0,1} + n_i^{1,0} + n_i^{1,1}}$, $\hat{p}_{Y,a}^i = \frac{n_i^{1,a}}{n_i^{0,a} + n_i^{1,a}}$, $\hat{q}_{Y,a}^i = 1 - \hat{p}_{Y,a}^i$ to calculate the empirical error. We will further present a theorem to show that we can achieve a desirable accuracy using the estimated values in Section 4.

At the end of this section, we provide a concise summary of our algorithm in Algorithm 1. It is worth noting that while in our experiment, we assume that $\pi_i$ is proportional to $n_i$, we may not know the exact values of $\pi_i$ in real applications. To enhance the robustness of our approach in such real-world scenarios, one can consider introducing a hypothesis space denoted as $H(\pi)$ to model the range of $\pi$ and incorporate $\max_{\pi \in H(\pi)}$ into Equations equation 8 and equation 9. In the experiments, we use the full sample version for data efficiency and still observe fairness control empirically

## 3.4   Extension to Equalized Odds

We now extend FedFaiREE from EOO to Equalized Odds (EO), which requires simultaneously controlling both the true positive rate and the false positive rate across groups.

Under EO, fairness constraints involve two types of conditional distributions (for $Y = 1$ and $Y = 0$). Similar to the EOO case, these quantities can be characterized by the relative positions (i.e., ranks or quantiles) of score thresholds within each group. Therefore, EO can be enforced by jointly aligning order statistics for both $Y = 1$ and $Y = 0$ across groups. For notational simplicity, let $n_i^{y,a}$ denote the number of samples in client $i$ with label $Y = y$ and attribute $A = a$. The corresponding rank variables $k^{y,a}$ and $\hat{k}_i^{y,a}$ are defined analogously to the EOO case.

We now present the following extension of Proposition 3.4.

**Proposition 3.6.** *Under Assumption 3.1, for $a \in \{0,1\}$, consider $k^{1,a} \in \{1,\ldots,n^{1,a}\}$, the corresponding $\hat{k}_i^{1,a}$ for $i \in [S]$ which are $\varepsilon$-approximate ranks and the score-based classifier $\phi(x,a) = \mathbb{1}\{f(x,a) > t_{(k^{1,a})}^{1,a}\}$ . Define*

$$
h_{y,a}(\boldsymbol{u}, \boldsymbol{v}) = \mathbb{P}\big( \sum_{i=1}^{S} \pi_i^{y,a} Q\left(u_i, n_i^{y,a} + 1 - u_i\right) - \sum_{i=1}^{S} \pi_i^{y,1-a} Q\left(v_i, n_i^{y,1-a} + 1 - v_i\right) \geq \alpha \big).
$$

*Then we have:*

$$
\mathbb{P}(|DEO(\phi)| \preceq (\alpha, \alpha)) \geq 1 - h_{1,1}^* - h_{1,0}^* - h_{0,1}^* - h_{0,0}^* \tag{11}
$$

*where the definitions of $M_i^{y,a}$, $m_i^{y,a}$, $\pi_i^{y,a}$, $Q(A,B)$ are similar to Proposition 3.4, $h_{1,1}^* = h_{y,a}(\boldsymbol{M}^{y,a}, \boldsymbol{m}^{y,a})$.*

Compared to the EOO case, the EO constraint introduces two additional terms corresponding to the $Y = 0$ population. As a result, fairness control requires simultaneous alignment of order statistics across both positive and negative classes, leading to a union bound over four terms.

Building upon Proposition 3.6, we construct the candidate set under EO constraints and apply the same rank-selection framework as before.

---

**Algorithm 2** FedFaiREE for EO

---

**Input:** Calibration Dataset $D_i = D_i^{0,0} \cup D_i^{0,1} \cup D_i^{1,0} \cup D_i^{1,1}$; pre-trained classifier $\phi_0$ with function f; fairness constraint parameter $\alpha$ ; Confidence level parameter $\beta$; Weights of different clients $\pi$

**Output:** classifier $\hat{\phi}(x,a) = \mathbb{1}\{f(x,a) > t_{(k^{1,a})}^{1,a}\}$

1: **Client Side:**
2: **for** i=1,2,..,$S$ **do**
3:     Score on train data points in $D_i$ and get $T_i^{y,a} = \{t_{i,1}^{y,a}, t_{i,2}^{y,a}, \cdots, t_{i,n_i^{y,a}}^{y,a}\}$
4:     Sort $T_i^{y,a}$
5:     Calculate q-digest of $T_i^{y,a}$ on client $i$
6:     Update digest to server
7: **end for**
8: **Server Side:**
9: Construct $K$ by $K = \{(k^{1,0}, k^{1,1}) | L(\boldsymbol{k}^{1,0}, \boldsymbol{k}^{1,1}) < 1 - \beta\}$, where $L = h_{1,1}^* + h_{1,0}^* + h_{0,1}^* + h_{0,0}^*$ corresponding to the right hand side of equation 11
10: Select optimal $(k_0, k_1)$ by minimizing equation 9 using estimated values $\hat{p}_a^i = \frac{n_i^{0,a} + n_i^{1,a}}{n_i^{0,0} + n_i^{0,1} + n_i^{1,0} + n_i^{1,1}}$ and $\hat{p}_{Y,a}^i = \frac{n_i^{1,a}}{n_i^{0,a} + n_i^{1,a}}$

---

# 4 Theoretical Guarantees

In this section, we establish theoretical guarantees for FedFaiREE, showing that it achieves both *finite-sample and distribution-free fairness control* and *near-optimal accuracy*.

At a high level, our results show that: 1). FedFaiREE satisfies the target fairness constraint with high probability in a distribution-free manner; and 2). the resulting classifier achieves a misclassification error close to that of the fair Bayes-optimal classifier, up to controllable approximation and estimation errors.

To derive these results, we first introduce a mild regularity condition on the score function.

**Assumption 4.1.** The distribution of $f(x, a)$ exhibits the following property. When conducting $N$ independent samplings to form a sample set, let $q_0$ be the $\beta$-quantile of the sample set. There exist function $\delta : \mathbb{N} \to \mathbb{R}$, constant $\gamma > 0$, such that $\lim_{N \to \infty} \delta(N) = 0$ and with a probability of at least $1 - \delta(N)$, for any $q$ considered as an $\varepsilon$-approximate $\beta$-quantile of the sample set, it satisfies that , $q$ lies within the $\gamma\varepsilon$-neighborhood of $q_0$.

Assumption 4.1 ensures that approximate quantiles obtained via distributed rank estimation are stable, in the sense that small rank errors translate into controlled perturbations of the corresponding thresholds. This condition is analogous to a Lipschitz-type regularity of the score distribution near the quantile of interest, and is standard in quantile-based analysis.

In the following theorem, we establish a theoretical basis for the accuracy of FedFaiREE. To facilitate comparison, we introduce the notion of the *fair Bayes-optimal classifier*, defined as the classifier achieving the lowest misclassification error under the fairness constraint. The formal definition is given in Lemma A.2. We denote the standard (unconstrained) Bayes-optimal classifier by

$$\phi^*(x, a) = \mathbb{1}\{f^*(x, a) > 1/2\},$$

where $f^* \in \arg\min_f \mathbb{P}(Y \neq \mathbb{1}\{f(x, a) > 1/2\})$. We now present the theoretical guarantees of Algorithm 1 proposed in the last section.

**Theorem 4.2.** *Under Assumptions 3.1 and 4.1, let $\alpha' < \alpha$, and let $\hat{\phi}$ be the output of Fed-FaiREE. Then:*

*(1) Fairness guarantee.*

$$|DEOO(\hat{\phi})| < \alpha$$

*holds with probability $(1 - \delta)^N$, where $N$ is the size of the candidate set.*

*(2) Accuracy guarantee. Suppose the density of $f^*$ under $A = a, Y = 1$ is continuous. If the input classifier satisfies $|f(x, a) - f^*(x, a)| \leq \epsilon_0$, then for any $\epsilon > 0$ such that $F^*_{(+)}(\epsilon + \gamma\varepsilon) \leq \frac{\alpha - \alpha'}{2} - F^*_{(+)}(2\epsilon_0)$, we have*

$$\mathbb{P}(\hat{\phi}(x, a) \neq Y) - \mathbb{P}(\phi^*_{\alpha'}(x, a) \neq Y) \leq 2F^*_{(+)}(2\epsilon_0) + 2F^*_{(+)}(\epsilon + \gamma\varepsilon) + 8\epsilon^2 + 20\epsilon + 2\theta, \tag{12}$$

*with probability at least $1 - 4\sum_{a=0}^{1}\sum_{i=1}^{S} e^{-2n_i^{0,a}\epsilon^2} - \prod_{i=1}^{S}(1 - F_{i(-)}^{1,0}(2\epsilon))^{n_i^{1,0}} - \prod_{i=1}^{S}(1 - F_{i(-)}^{1,1}(2\epsilon))^{n_i^{1,1}} - \delta$, where $\delta = \delta^{1,0}(n^{1,0}) + \delta^{1,1}(n^{1,1})$, $\theta$ is defined in Proposition 3.5 and the definition of $F_{(+)}$ and $F_{(-)}$ are shown in Lemma A.4.*

The excess risk bound consists of three components: 1). approximation error $F_{(+)}^*(2\epsilon_0)$, which reflects how close the input score function $f$ is to the Bayes-optimal $f^*$; 2). quantile estimation error $F_{(+)}^*(\epsilon + \gamma\varepsilon)$, which arises from approximate rank estimation and is controlled by $\varepsilon$; and 3). statistical error terms (in $\epsilon$ and $\theta$), which vanish as sample sizes increase. Together, these terms show that FedFaiREE achieves near-optimal accuracy with DEOO constraints, provided that the initial score function is sufficiently accurate and the quantile approximation error is controlled, underscoring the effectiveness of our approach in minimizing errors when ensuring fairness in a distribution-free and small-sample manner.

Similarly, we provide theoretical guarantees for DEO fairness.

**Theorem 4.3.** *Under Assumptions 3.1 and 4.1, given $\alpha' < \alpha$. Suppose $\hat{\phi}$ is the final output of FedFaiREE with target DEO constraint. We then have:*

*(1) Fairness guarantee.*
$$|DEO(\hat{\phi})| < \alpha$$

*holds with probability $(1-\delta)^N$, where $N$ is the size of the candidate set.*

*(2) Accuracy guarantee. Suppose the density distribution functions of $f^*$ under $A = a, Y = 1$ are continuous. When the input classifier $f$ satisfies $|f(x,a) - f^*(x,a)| \le \epsilon_0$, for any $\epsilon > 0$ such that $F_{(+)}^*(\epsilon + \gamma\varepsilon) \le \frac{\alpha - \alpha'}{2} - F_{(+)}^*(2\epsilon_0)$, we have*

$$\mathbb{P}(\hat{\phi}(x,a) \neq Y) - \mathbb{P}(\phi_{\alpha'}^*(x,a) \neq Y) \le 2F_{(+)}^*(2\epsilon_0) + 2F_{(+)}^*(\epsilon + \gamma\varepsilon) + 2\theta + O(\epsilon) \tag{13}$$

*with probability $1 - 4\sum_{a=0}^1 \sum_{i=1}^S e^{-2n_i^{0,a}\epsilon^2} - \prod_{i=1}^S \left(1 - F_{i(-)}^{1,0}(2\epsilon)\right)^{n_i^{1,0}} - \prod_{i=1}^S \left(1 - F_{i(-)}^{1,1}(2\epsilon)\right)^{n_i^{1,1}} - \delta$, where the definitions of $\delta$, $\theta$, $F_{(+)}$, $F_{(-)}$ are same as Theorem 4.2.*

Compared to the DEOO case, the DEO result involves both positive and negative classes, but the overall structure of the bound remains similar. This shows that our framework extends naturally to stronger fairness notions without degrading statistical guarantees.

## 5 Numerical Experiments

In this section, we empirically evaluate FedFaiREE on multiple benchmark datasets under both semi-synthetic and real heterogeneous settings, and compare it with existing baselines in terms of the fairness–accuracy trade-off.

### 5.1 Numerical results for EOO

We evaluate FedFaiREE on two semi-synthetic datasets, Adult (Dua et al., 2017) and Compas (Dieterich et al., 2016), where decentralized data are generated following the pipeline of Ezzeldin et al. (2023), as well as two real-world datasets, ACSIncome (Ding et al., 2021) and CelebA (Liu et al., 2015), which naturally exhibit heterogeneous federated structures.

We apply FaiREE and FedFaiREE as post-processing methods on top of standard federated learning baselines, including FedAvg (McMahan et al., 2017), FedFB (Zeng et al., 2021), and FairFed (Ezzeldin et al., 2023). All models are trained using two-layer neural networks, except for CelebA where we adopt ResNet18. Further implementation details are provided in Appendix C.

### 5.1.1 Semi-synthetic Data

Adult dataset (Dua et al., 2017), which is employed for the prediction task that determines whether an individual's income exceeds \$50,000, comprises 45,222 samples, featuring various attributes including age, education, and more. Compas dataset (Dieterich et al., 2016), whose task is to predict whether a person will conduct crime in the future, comprises 7214 samples. Gender is chosen as the sensitive feature for both datasets.

**Data Processing.** To realize the decentralized conditions and account for heterogeneity across clients, we adopt the approach introduced by Ezzeldin et al. (2023). Specifically, we initiated the process by randomly sampling proportions for various sensitive attributes within each client, using the Dirichlet distribution. Subsequently, we partitioned the dataset into client-specific subsets based on these proportions. Within each of these subsets, we performed an 80-20 split, allocating 80% of the data as the local client training set and reserving the remaining 20% for the test set. For the numerical experiments, we repeated this procedure 100 times on both Adult and Compas datasets.

**Result and Analysis.** Table 1 summarizes the performance on the Adult and Compas datasets. We also revisit Figure 1 here. The distribution of $|DEOO|$ on the Adult dataset shows that FedFaiREE shifts the fairness disparity toward smaller values and better controls the upper tail compared with FedAvg, FairFed, and centralized FaiREE. The dashed vertical line indicates the target tolerance $\alpha = 0.10$. Across all backbone models (FedAvg, FedFB, FairFed), FedFaiREE consistently achieves substantially lower $|DEOO|$ after post-processing, compared to both the original models and post-processing by FaiREE. More importantly, the empirical $|DEOO|_{95}$ aligns closely with the target confidence level $\beta = 0.95$, demonstrating that our high-probability fairness guarantee is effective in practice. In addition, on Adult, while FaiREE improves fairness in centralized settings, its performance degrades under heterogeneous client distributions. In contrast, FedFaiREE explicitly accounts for client-level variability via local order statistics, leading to stable fairness control across all settings. Further, despite enforcing strict fairness constraints, FedFaiREE maintains accuracy comparable to the baselines. This validates our theoretical result that fairness can be achieved with minimal loss in predictive performance. Compared with Adult, Compas has a smaller sample size, and therefore we also partition it into fewer clients. In this setting, the thresholds selected by FaiREE and FedFaiREE are nearly identical, which leads to the similar performance in Table 1. Overall, these results confirm that FedFaiREE effectively balances fairness and accuracy under finite-sample and heterogeneous federated settings. More numerical experiments with varying values of $\alpha$ and $\beta$, as well as additional ablation studies are deferred to Appendix C.

Table 1: **Results on Adult and Compas dataset.** We conducted 100 experimental repetitions for each model on both datasets and compared the accuracy and fairness indicators of different models. The Method and $\alpha$ columns indicate whether FedFaiREE or FaiREE was used or not and the fairness constraint. Confidence level $\beta$ is set to be 95% throughout the experiments. $\overline{ACC}$ and $\overline{|DEOO|}$ represent the averages of accuracy and DEOO (defined in equation 2). $|DEOO|_{95}$ represents the 95% quantile of DEOO since we set the confidence level of FedFaiREE to 95% in our experiments.

| Model | Method | | Adult | | | | Compas | | |
|---|---|---|---|---|---|---|---|---|---|
| | | $\alpha$ | $\overline{ACC}$ | $\overline{|DEOO|}$ | $|DEOO|_{95}$ | $\alpha$ | $\overline{ACC}$ | $\overline{|DEOO|}$ | $|DEOO|_{95}$ |
| **FedAvg** | / | / | 0.844 | 0.131 | 0.178 | / | 0.662 | 0.126 | 0.223 |
| | FaiREE | 0.10 | 0.844 | 0.071 | 0.128 | 0.15 | 0.659 | **0.051** | **0.137** |
| | FedFaiREE | 0.10 | 0.843 | **0.038** | **0.083** | 0.15 | 0.659 | **0.051** | **0.137** |
| **FedFB** | / | / | 0.850 | 0.057 | 0.117 | / | 0.642 | 0.107 | 0.174 |
| | FaiREE | 0.10 | 0.850 | 0.055 | 0.109 | 0.15 | 0.641 | **0.062** | **0.125** |
| | FedFaiREE | 0.10 | 0.850 | **0.036** | **0.083** | 0.15 | 0.641 | **0.062** | **0.125** |
| **FairFed** | / | / | 0.842 | 0.069 | 0.118 | / | 0.648 | 0.097 | 0.166 |
| | FaiREE | 0.10 | 0.842 | 0.066 | 0.112 | 0.15 | 0.645 | **0.047** | **0.114** |
| | FedFaiREE | 0.10 | 0.841 | **0.037** | **0.081** | 0.15 | 0.645 | **0.047** | **0.114** |

### 5.1.2 Real Data Analysis

To validate the effectiveness of FedFaiREE in scenarios with naturally heterogeneous distributions, we further consider the ACSIncome dataset (Ding et al., 2021) and CelebA dataset (Liu et al., 2015). In the ACSIncome dataset, the task is to predict whether an individual's income is above $50,000, with the sensitive label being Race (white/non-white), and the data partitioned across 50 states. CelebA dataset is a large-scale face attributes dataset comprising more than 200k images of 10k target celebrities, each annotated with 40 attributes. We create clients by grouping every 20 celebrities together, resulting in 508 clients. Following prior work (Park et al., 2022), we use Attractive as target variable and Male as sensitive attribute.

Table 2 reports the results on ACSIncome and CelebA, which exhibit natural heterogeneity. We observe that FedFaiREE significantly reduces $|DEOO|$ across both datasets while maintaining competitive accuracy. Notably, the improvement is particularly pronounced on CelebA, where data heterogeneity is substantial due to the large number of clients. This demonstrates that the rank-based formulation of FedFaiREE is especially effective in realistic federated environments, where distributional differences across clients are inherent and cannot be ignored. Detailed hyperparameter selection and experimental set-up are provided in Sections C.2 and C.3 of the Appendix respectively.

Table 2: **Results on ACSIncome and CelebA datasets.**

| Model | FedFaiREE | | ACSIncome | | | CelebA | |
|---|---|---|---|---|---|---|---|
| | | $\alpha$ | $\overline{ACC}$ | $\overline{|DEOO|}$ | $\alpha$ | $\overline{ACC}$ | $\overline{|DEOO|}$ |
| **FedAvg** | ✗ | / | 0.808 | 0.126 | / | 0.709 | 0.280 |
| | ✓ | 0.10 | 0.806 | **0.041** | 0.15 | 0.684 | **0.099** |
| **FairFed** | ✗ | / | 0.773 | 0.092 | / | 0.721 | 0.324 |
| | ✓ | 0.10 | 0.771 | **0.044** | 0.15 | 0.697 | **0.086** |

## 5.2 Numerical results for EO

We evaluate FedFaiREE under the Equalized Odds (DEO) constraint, as introduced in Section 3.4. Since Equalized Odds requires controlling both the true positive rate and the false positive rate across groups, we report DEOO and DPE, where

$$\text{DPE} = \mathbb{P}_{X|A=1,Y=0}(\widehat{Y}=1) - \mathbb{P}_{X|A=0,Y=0}(\widehat{Y}=1).$$

The results are summarized in Tables 3 and 4. We observe that FedFaiREE consistently improves both DEOO and DPE across all models. Importantly, these fairness gains are achieved while maintaining comparable accuracy, demonstrating that our method effectively handles the stricter DEO constraint without significant loss in predictive performance.

Moreover, the improvements are consistent across different backbone methods (FedAvg, FedFB, and FairFed), indicating that the proposed rank-based framework is robust and broadly applicable in federated settings.

Table 3: **Results of FedFaiREE for DEO on Adult dataset.** We conducted 100 experimental repetitions for each model on both datasets and compared the accuracy and fairness indicators of different models. The "FedFaiREE" and "$\alpha$" columns indicate whether FedFaiREE was used or not. "$\overline{ACC}$", "$\overline{|DEOO|}$" and "$\overline{|DPE|}$" represent the averages of accuracy, DEOO (defined in equation 2) and DPE (defined in equation 35), respectively. "$|DEOO|_{95}$" and "$|DPE|_{95}$" represent the 95% quantile of DEOO and DPE since we set the confidence level of FedFaiREE to 95% in our experiments.

| Model | FedFaiREE | $\alpha$ | $\overline{ACC}$ | $\overline{|DEOO|}$ | $|DEOO|_{95}$ | $\overline{|DPE|}$ | $|DPE|_{95}$ |
|---|---|---|---|---|---|---|---|
| | | | | | Adult | | |
| **FedAvg** | ✗ | / | 0.844 (0.003) | 0.131 (0.030) | 0.178 | 0.088 (0.005) | 0.097 |
| | ✓ | 0.10 | 0.843 (0.003) | **0.037** (0.025) | **0.082** | **0.064** (0.007) | **0.075** |
| **FedFB** | ✗ | / | 0.850 (0.003) | 0.057 (0.034) | 0.117 | 0.066 (0.007) | 0.077 |
| | ✓ | 0.10 | 0.850 (0.003) | **0.036** (0.025) | **0.083** | **0.061** (0.006) | **0.070** |
| **FairFed** | ✗ | / | 0.842 (0.003) | 0.069 (0.034) | 0.118 | 0.072 (0.006) | 0.083 |
| | ✓ | 0.10 | 0.841 (0.003) | **0.037** (0.026) | **0.081** | **0.063** (0.006) | **0.071** |

Table 4: **Results of FedFaiREE for DEO on Compas dataset.** We conducted 100 experimental repetitions for each model on both datasets and compared the accuracy and fairness indicators of different models. All notations are the same as in Table 3

| Model | FedFaiREE | $\alpha$ | $\overline{ACC}$ | $\overline{|DEOO|}$ | $|DEOO|_{95}$ | $\overline{|DPE|}$ | $|DPE|_{95}$ |
|---|---|---|---|---|---|---|---|
| | | | | | Compas | | |
| **FedAvg** | ✗ | / | 0.662 (0.011) | 0.126 (0.056) | 0.223 | 0.083 (0.032) | 0.136 |
| | ✓ | 0.15 | 0.652 (0.036) | **0.049** (0.045) | **0.137** | **0.028** (0.024) | **0.072** |
| **FedFB** | ✗ | / | 0.642 (0.011) | 0.107 (0.043) | 0.174 | 0.066 (0.028) | 0.112 |
| | ✓ | 0.15 | 0.642 (0.010) | **0.062** (0.040) | **0.125** | **0.036** (0.024) | **0.081** |
| **FairFed** | ✗ | / | 0.648 (0.011) | 0.097 (0.047) | 0.166 | 0.087 (0.036) | 0.148 |
| | ✓ | 0.15 | 0.642 (0.029) | **0.047** (0.036) | **0.114** | **0.037** (0.028) | **0.085** |

# 6 Extension

In this section, we extend FedFaiREE to several practically relevant settings, including label shift at test time and multi-group fairness. These extensions demonstrate the flexibility of our rank-based framework and its applicability beyond the basic binary-group setting.

## 6.1 Label Shift in Test Set

We first consider the label shift setting, where the test distribution differs from the training distribution in class proportions. This scenario is common in real-world deployments (Plassier et al., 2023; Tian et al., 2023), where the marginal distribution of labels may evolve over time while the conditional distribution $P(X, A \mid Y)$ remains stable.

To accommodate this setting, we introduce the counterpart of Assumption 3.1 in the following assumption.

**Assumption 6.1.** The training data points on client $i$ are i.i.d drawn from the distribution $P_i$, and we further assume the global distribution $P$ is a mixture of $P_1, \cdots, P_S$ with weight $\{\pi_i\}_{i \in [S]} \in \Delta_S$, while the test data points are sampled from another distribution $P_{S+1}$, heterogeneity between $P$ and which induced due to label shift, that is, we assume that

$$P^{mix} = \sum_{i=1}^{S} \pi_i P_i = P(X, A|Y) * P^{mix}(Y), \ \left(X^{\text{test}}, Y^{\text{test}}\right) \sim P_{S+1} = P(X, A|Y) * P_{S+1}(Y). \ (14)$$

This assumption captures the classical label-shift setting, in which only the class proportions change between training and testing, while the conditional distribution of covariates and sensitive attributes given the label remains stable. We adapt FedFaiREE to this setting by modifying the empirical objective used for rank selection. Specifically, we replace equation 9 with a reweighted error estimator that accounts for the discrepancy between training and test label distributions:

$$
\begin{aligned}
\hat{\mathbb{P}}\left(\hat{\phi}(x, a) \neq Y\right) = \sum_{i=1}^{S} \pi_i \big[ & \frac{\hat{k}_i^{1,0} + 0.5}{n_i^{1,0} + 1} p_0^i p_{Y,0}^i w^{1,0} + \frac{\hat{k}_i^{1,1} + 0.5}{n_i^{1,1} + 1} p_1^i p_{Y,1}^i w^{1,1} \\
& + \frac{n_i^{0,0} + 0.5 - \hat{k}_i^{0,0}}{n_i^{0,0} + 1} p_0^i q_{Y,0}^i w^{0,0} + \frac{n_i^{0,1} + 0.5 - \hat{k}_i^{0,1}}{n_i^{0,1} + 1} p_1^i q_{Y,1}^i w^{0,1} \big],
\end{aligned}
\tag{15}
$$

where $w^{y,a} = \frac{p_a^{S+1} p_{Y,a}^{S+1}}{p_a p_{Y,a}}$. The weights $w^{y,a}$ reweight each subgroup to reflect its prevalence under the test distribution, effectively correcting the bias introduced by label shift. As a result, similar to Proposition 3.2, we have the following proposition.

**Proposition 6.2.** *Under Assumption 6.1, the misclassification error can be estimated by equation 15. Further, discrepancy between empirical error and true error is limited by following inequality:*

$$\left|\mathbb{P}\left(\hat{\phi}(x, a) \neq Y\right) - \hat{\mathbb{P}}\left(\hat{\phi}(x, a) \neq Y\right)\right| \leq \theta' \tag{16}$$

---

**Algorithm 3** FedFaiREE for label shift case

---

**Input:** Calibration Dataset $D_i = D_i^{0,0} \cup D_i^{0,1} \cup D_i^{1,0} \cup D_i^{1,1}$; pre-trained classifier $\phi_0$ with function f; fairness constraint parameter $\alpha$ ; Confidence level parameter $\beta$; Weights of different clients $\pi$

**Output:** classifier $\hat{\phi}(x,a) = \mathbb{1}\{f(x,a) > t^{1,a}_{(k^{1,a})}\}$

 1: **Client Side:**
 2: **for** i=1,2,..,$S$ **do**
 3:    Score on train data points in $D_i$ and get $T_i^{y,a} = \{t^{y,a}_{i,1}, t^{y,a}_{i,2}, \cdots, t^{y,a}_{i,n_i^{y,a}}\}$
 4:    Sort $T_i^{y,a}$
 5:    Calculate q-digest of $T_i^{y,a}$ on client $i$
 6:    Update digest to server
 7: **end for**
 8: **Server Side:**
 9: Construct $K$ by $K = \{(k^{1,0}, k^{1,1}) | L(\boldsymbol{k}^{1,0}, \boldsymbol{k}^{1,1}) < 1 - \beta\}$
10: Select optimal $(k_0, k_1)$ by minimizing equation 15 using estimated values $\hat{p}_a^i = \frac{n_i^{0,a} + n_i^{1,a}}{n_i^{0,0} + n_i^{0,1} + n_i^{1,0} + n_i^{1,1}}$ and $\hat{p}^i_{Y,a} = \frac{n_i^{1,a}}{n_i^{0,a} + n_i^{1,a}}$

---

where $e_i^{y,a} = \frac{2\lfloor \varepsilon n_i^{y,a} \rfloor + 1}{2(n_i^{y,a} + 1)}$ and

$$\theta' = \sum_{i=1}^{S} \pi_i \left[ e_i^{0,0} p_0^i q_{Y,0}^i w^{0,0} + e_i^{0,1} w^{0,1} p_0^i p_{Y,0}^i + e_i^{1,0} w^{1,0} p_1^i q_{Y,1}^i + e_i^{1,1} w^{1,1} p_1^i p_{Y,1}^i \right].$$

Proposition 6.2 shows that the reweighted estimator consistently approximates the test-time misclassification error, despite the distribution shift. This result enables the server to select candidate classifiers based on an empirical objective that accurately tracks test-time performance. The overall procedure is summarized in Algorithm 3. We further establish fairness and accuracy guarantees analogous to Theorem 4.2.

**Theorem 6.3.** *Under Assumptions 4.1 and 6.1, given $\alpha' < \alpha$. Suppose $\hat{\phi}$ is the final output of FedFaiREE. We then have:*

*(1) Fairness guarantee.*

$$|DEOO(\hat{\phi})| < \alpha$$

*holds with probability $(1 - \delta)^N$, where $N$ is the size of the candidate set.*

*(2) Accuracy guarantee. Under the assumptions specified in Appendix A.5, with high probability we have*

$$\mathbb{P}(\hat{\phi}(x,a) \neq Y) - \mathbb{P}(\phi^*_{\alpha'}(x,a) \neq Y) \leq 2F^*_{(+)}(2\epsilon_0) + 2F^*_{(+)}(\epsilon + \gamma\varepsilon) + 2\theta' + O(\epsilon), \tag{17}$$

Theorem 6.3 shows that FedFaiREE retains both fairness control and near-optimal accuracy under label shift. This is particularly important in deployment scenarios where label distributions may change over time.

## 6.2   Extension to the Multi-Group Setting

We now extend FedFaiREE to the multi-group setting, where the sensitive attribute takes values in $\mathcal{A} = \{0, 1, \ldots, A_0\}$ with $A_0 > 1$. This setting is common in practice, where fairness must

be enforced across multiple demographic groups. In such settings, the fairness constraint must ensure that the classifier behaves similarly across all protected groups, rather than only between two groups.

We begin by introducing the notion of Equality of Opportunity in the multi-group setting.

**Definition 6.4.** (Equality of Opportunity; Multiple Groups) A classifier satisfies Equality of Opportunity if the true positive rate is equal across all protected groups:

$$\mathbb{P}_{X|A=0,Y=1}(\widehat{Y} = 1) = \mathbb{P}_{X|A=a,Y=1}(\widehat{Y} = 1), \quad \forall a \in \mathcal{A}.$$

This definition generalizes the binary-group notion of Equality of Opportunity by requiring parity of true positive rates across all groups.

To quantify deviations from this condition, we define

$$DEOOM = \max_a \left| \mathbb{P}_{X|A=a,Y=1}(\widehat{Y} = 1) - \mathbb{P}_{X|A=0,Y=1}(\widehat{Y} = 1) \right|.$$

Thus, controlling $DEOOM$ amounts to ensuring that no protected group differs too much from the reference group in terms of true positive rate. We now extend the fairness control result underlying FedFaiREE to this multi-group setting.

**Proposition 6.5.** *Under Assumption 3.1, for $a \in \{0, 1, \cdots, A_0\}$, consider $k^{1,a} \in \{1, \ldots, n^{1,a}\}$, the corresponding $\hat{k}_i^{1,a}$ for $i \in [S]$ which are $\varepsilon$-approximate ranks and the score-based classifier $\phi(x, a) = \mathbb{1}\{f(x, a) > t_{(k^{1,a})}^{1,a}\}$ . Define*

$$
\begin{aligned}
h_{y,a}^* =& \mathbb{P}\left( \sum_{i=1}^S \pi_i^{y,a} Q\left(M_i^{1,a}, n_i^{y,a} + 1 - M_i^{1,a}\right) - \sum_{i=1}^S \pi_i^{y,0} Q\left(m_i^{1,0}, n_i^{y,0} + 1 - m_i^{1,0}\right) \geq \alpha \right) \\
&+ \mathbb{P}\left( \sum_{i=1}^S \pi_i^{y,0} Q\left(M_i^{1,0}, n_i^{y,0} + 1 - M_i^{1,0}\right) - \sum_{i=1}^S \pi_i^{y,a} Q\left(m_i^{1,a}, n_i^{y,a} + 1 - m_i^{1,a}\right) \geq \alpha \right)
\end{aligned}
$$

*Then we have:*

$$\mathbb{P}(|DEOOM(\phi)| > \alpha) \leq \sum_{a=1}^{A_0} h_{1,a}^* \tag{18}$$

*where $\pi_i^{1,a}$, $\pi_i^{1,0}$ are similarly defined as in Proposition 3.4. $M_i^{1,a} = \min\left(\lceil \hat{k}_i^{1,a} + \varepsilon n_i^{1,a} \rceil, n_i^{1,a} + 1\right)$, $m_i^{1,a} = \max\left(\lceil \hat{k}_i^{1,a} - \varepsilon n_i^{1,a} \rceil, 0\right)$, $M_i^{1,0}$ and $m_i^{1,0}$ are similarly defined. $Q(\alpha, \beta)$ are independent random variables and $Q(\alpha, \beta) \sim Beta(\alpha, \beta)$. Especially, we define $Q(0, \beta) = 0$ and $Q(\alpha, 0) = 1$ for $\alpha, \beta \neq 0$.*

Proposition 6.5 is the multi-group analogue of Proposition 3.4. It shows that the probability of violating the fairness constraint can still be controlled by aggregating client-level uncertainty through Beta-distributed order-statistics bounds. Compared to the binary-group case, fairness control now requires bounding deviations for each group relative to the reference group. This leads to a summation over all groups in equation 18, reflecting a union bound over group-wise fairness violations. Next, we extend the error estimation result to the multi-group setting.

**Algorithm 4** FedFaiREE for Multi-Groups

---

**Input:** Calibration Dataset $D_i = D_i^{0,0} \cup D_i^{0,1} \cup D_i^{1,0} \cup D_i^{1,1}$; pre-trained classifier $\phi_0$ with function f; fairness constraint parameter $\alpha$ ; Confidence level parameter $\beta$; Weights of different clients $\pi$

**Output:** classifier $\hat{\phi}(x,a) = \mathbb{1}\{f(x,a) > t_{(k^{1,a})}^{1,a}\}$

1: **Client Side:**
2: **for** i=1,2,..,$S$ **do**
3:     Score on train data points in $D_i$ and get $T_i^{y,a} = \{t_{i,1}^{y,a}, t_{i,2}^{y,a}, \cdots, t_{i,n_i^{y,a}}^{y,a}\}$
4:     Sort $T_i^{y,a}$
5:     Calculate q-digest of $T_i^{y,a}$ on client $i$
6:     Update digest to server
7: **end for**
8: **Server Side:**
9: Construct $K$ by $K = \{(k^{1,0}, k^{1,1}, \cdots, k^{1,A_0}) | L < 1 - \beta\}$, where L is defined by the right-hand side of Inequality 18
10: Select optimal $(k^{1,0}, k^{1,1}, \cdots, k^{1,A_0})$ by minimizing equation 19 using estimated values $\hat{p}_a^i$ and $\hat{p}_{Y,a}^i$

---

**Proposition 6.6.** *Under Assumption 3.1, the misclassification error can be estimated by*

$$\hat{\mathbb{P}}\left(\hat{\phi}(x,a) \neq Y\right) = \sum_{i=1}^{S}\left[\pi_i \sum_{a=0}^{A_0}\left(\frac{\hat{k}_i^{1,a} + 0.5}{n_i^{1,a} + 1}p_a^i p_{Y,a}^i + \frac{n_i^{0,a} + 0.5 - \hat{k}_i^{0,a}}{n_i^{0,a} + 1}p_a^i q_{Y,a}^i\right)\right] \tag{19}$$

*Further, the discrepancy between empirical error and true error is upper bounded by the following:*

$$\left|\mathbb{P}\left(\hat{\phi}(x,a) \neq Y\right) - \hat{\mathbb{P}}\left(\hat{\phi}(x,a) \neq Y\right)\right| \leq \theta, \tag{20}$$

*where $\theta = \sum_{i=1}^{S}\left[\pi_i \sum_{a=0}^{A_0}\left(e_i^{0,a}p_a^i q_{Y,a}^i + e_i^{1,a}p_1^i q_{Y,a}^i\right)\right]$, $e_i^{y,a} = \frac{2\lfloor \varepsilon n_i^{y,a}\rfloor + 1}{2(n_i^{y,a}+1)}$.*

Proposition 6.6 shows that, the error estimator retains the same structure as in the binary case, with an additional summation over groups. This shows that the rank-based formulation scales naturally with the number of groups.

Based on Propositions 6.5 and 6.6, we obtain the following extension of FedFaiREE, summarized in Algorithm 4. We next state the corresponding theoretical guarantee.

**Theorem 6.7.** *Under Assumption 3.1 and 4.1, given $\alpha' < \alpha$. Suppose $\hat{\phi}$ is the final output of FedFaiREE, we have:*

*(1) Fairness guarantee.*

$$|DEOOM(\hat{\phi})| < \alpha$$

*holds with probability $(1-\delta)^N$, where N is the size of the candidate set.*

*(2) Accuracy guarantee. Suppose the density distribution functions of $f^*$ under $A = a, Y = 1$ are continuous. When the input classifier $f$ satisfies $|f(x,a) - f^*(x,a)| \leq \epsilon_0$, for any $\epsilon > 0$ such that $F_{(+)}^*(\epsilon + \gamma\varepsilon) \leq \frac{\alpha - \alpha'}{2} - F_{(+)}^*(2\epsilon_0)$, we have*

$$\mathbb{P}(\hat{\phi}(x,a) \neq Y) - \mathbb{P}\left(\phi_{\alpha'}^*(x,a) \neq Y\right) \leq 2F_{(+)}^*(2\epsilon_0) + 2F_{(+)}^*(\epsilon + \gamma\varepsilon) + 2\theta + O(\epsilon) \tag{21}$$

*with probability at least $1 - 4\sum_{a=0}^{A_0}\sum_{i=1}^{S}e^{-2n_i^{0,a}\epsilon^2} - \sum_{a=0}^{A_0}\prod_{i=1}^{S}\left(1 - F_{i(-)}^{1,a}(2\epsilon)\right)^{n_i^{1,a}} - \delta$. Here $\delta =$*

$\sum_{a=0}^{A_0} \delta^{1,a}(n^{1,a})$, $\theta$ *is defined in Proposition 6.6 and the definition of* $F_{(+)}$ *and* $F_{(-)}$ *are shown in Lemma A.4*

The theorem shows that the main guarantees of FedFaiREE extend seamlessly to the multi-group setting. In particular, the algorithm continues to enforce fairness with high probability, while achieving near-optimal accuracy with explicit finite-sample bounds.

Overall, these extensions highlight the generality of the proposed framework. FedFaiREE can accommodate distribution shift and complex fairness notions without altering its core rank-based structure. We defer the extension to multi-label settings to Appendix A.6.

# 7 Conclusion

In this paper, we introduce FedFaiREE, a small-sample and distribution-free approach to guarantee fairness constraints under the federated learning setting. FedFaiREE addresses concerns that commonly exist in federated learning, such as client heterogeneity, small samples, and limited communication costs. The FedFaiREE framework can be applied to a wide range of group fairness notions and various scenarios, including label shifts and multi-group settings.

For future work, an exploration of more efficient distributed quantile algorithms for rank and quantile calculations within the FedFaiREE framework could significantly enhance its scalability and performance. Moreover, exploring a broader range of application scenarios and assessing its performance in conjunction with in-processing fair federated learning frameworks could yield valuable insights.

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

**Appendix Organization.** The appendix is organized as follows. Appendix A presents the proofs of the main propositions and theorems, with Appendix A.1-A.5 devoted to the theoretical results in the main text and Section A.6 discussing the extension to the multi-label setting. Appendix B considers additional fairness notions beyond those studied in the main text, including Demographic Parity (Agarwal et al., 2018), Predictive Equality (Hardt et al., 2016), and Overall Accuracy Equality (Zafar et al., 2017), and develops the corresponding methodological extensions and theoretical guarantees. Appendix C provides further experimental details, including the choices of hyperparameters and optimizers, along with extensive ablation studies. These include analyses under different levels of client heterogeneity, fairness constraints $\alpha$, confidence levels $\beta$, and input scorer $f$ quality, as well as comparisons with local FaiREE, centralized FedFaiREE, and varying levels of client participation. Additional experimental results on more datasets are also reported there. Appendix D elaborates on the connections between our work, FaiREE, and other related literature. Appendix E discusses the treatment of tied and discrete score distributions through randomized tie-breaking, and clarifies the role of continuity assumptions in the finite-sample fairness guarantee.

Table 5: Summary of key notation.

| Notation | Meaning |
|---|---|
| $[S]$, $i$ | Set of clients $[S] = \{1, \ldots, S\}$ and client index. |
| $X, A, Y$ | Non-sensitive features, sensitive attribute, and binary label. |
| $P_i$, $P^{\mathrm{mix}}$, $\pi_i$ | Client $i$'s distribution, global mixture distribution $P^{\mathrm{mix}} = \sum_{i=1}^{S} \pi_i P_i$, and client mixture weight. |
| $\mathcal{D}_i$, $\mathcal{D}_i^{y,a}$, $n_i^{y,a}$ | Local dataset on client $i$, its subgroup with $Y = y, A = a$, and the corresponding subgroup sample size. |
| $p_a^i$, $p_{Y,a}^i$, $q_{Y,a}^i$ | Client-level group and label probabilities: $p_a^i = \mathrm{Pr}_i(A = a)$, $p_{Y,a}^i = \mathrm{Pr}_i(Y = 1 \mid A = a)$, and $q_{Y,a}^i = 1 - p_{Y,a}^i$. Hatted versions denote empirical estimates. |
| $f$, $\phi_0$, $\phi$ | Pre-trained score function, initial classifier $\phi_0(x, a) = \mathbf{1}\{f(x, a) > c\}$, and post-processed classifier $\phi(x, a) = \mathbf{1}\{f(x, a) > \lambda_a\}$. |
| $\lambda_a$ | Group-specific threshold for sensitive group $A = a$. |
| $t_{i,j}^{y,a}$, $T_i^{y,a}$ | Score of the $j$-th sample in $\mathcal{D}_i^{y,a}$, and the sorted local score set for subgroup $(y, a)$. |
| $t_{i,(r)}^{y,a}$, $t_{(r)}^{y,a}$ | The $r$-th local order statistic on client $i$, and the $r$-th global order statistic after pooling clients. |
| $k^{y,a}$ | Global rank of the selected threshold in the sorted global score set $T^{y,a}$. |
| $k_i^{y,a}$, $\widehat{k}_i^{y,a}$ | Exact local rank and $\varepsilon_{\mathrm{r}}$-approximate local rank of the selected threshold in $T_i^{y,a}$. |
| $\mathbf{k}^{y,a}$, $\widehat{\mathbf{k}}^{y,a}$ | Vectors of exact and approximate local ranks across clients. |
| $\alpha$, $\beta$ | Fairness tolerance and confidence-level parameter. |
| $DEOO(\phi)$ | Equality-of-Opportunity disparity. |
| $DEO(\phi)$ | Equalized-Odds disparity vector. |

# A   Proofs

To improve readability, we summarize the main notation and conventions used throughout the paper in Table 5.

## A.1 Proof for Proposition 3.2 and 3.4

We first introduce following lemma

**Lemma A.1.** *If $t_i^{y,a}$ is variable with continuous density function, we have*

$$F_i^{y,a}\left(t_{i,\left(k_i^{y,a}\right)}^{y,a}\right) \sim \text{Beta}\left(k_i^{y,a}, n_i^{y,a} - k_i^{y,a} + 1\right)$$

.

*Proof of Lemma A.1.* $F_i^{y,a}$ represents the continuous cumulative distribution functions of $t_i^{y,a}$, and thus we have $F_i^{y,a}(t_i^{y,a}) \sim U(0,1)$. Furthermore, as $F_i^{y,a}\left(t_{i,\left(k_i^{y,a}\right)}^{y,a}\right)$ denotes the $k_i^{y,a}$-th order statistic of $n_i^{y,a}$ i.i.d samples from $U(0,1)$, we can conclude that $F^{y,a}\left(t_{i,\left(k_i^{y,a}\right)}^{y,a}\right) \sim$ Beta $\left(k_i^{y,a}, n^{y,a} - k_i^{y,a} + 1\right)$ $\qquad \square$

Back to proof of the Proposition 3.2, the classifier is

*Proof of Proposition 3.2.*

$$\phi = \begin{cases} \mathbb{1}\left\{f(x,0) > t_{(k^{1,0})}^{1,0}\right\}, a = 0 \\ \mathbb{1}\left\{f(x,1) > t_{(k^{1,1})}^{1,1}\right\}, a = 1 \end{cases}$$

we have:

$$
\begin{aligned}
&\mathbb{P}(|DEOO(\phi)| > \alpha) \\
&= \mathbb{P}(|F^{1,1}(t_{(k^{1,1})}^{1,1}) - F^{1,0}(t_{(k^{1,0})}^{1,0})| > \alpha) \\
&= \mathbb{P}(\sum_{i=1}^{S} \pi_i^{1,1} F_i^{1,1}(t_{(k^{1,1})}^{1,1}) - \sum_{i=1}^{S} \pi_i^{1,0} F_i^{1,0}(t_{(k^{1,0})}^{1,0}) > \alpha) \\
&\quad + \mathbb{P}(\sum_{i=1}^{S} \pi_i^{1,1} F_i^{1,1}(t_{(k^{1,1})}^{1,1}) - \sum_{i=1}^{S} \pi_i^{1,0} F_i^{1,0}(t_{(k^{1,0})}^{1,0}) < -\alpha) \\
&\triangleq A + B
\end{aligned}
$$

So we only need to calculate $A$ and $B$ and It is easy to prove that we only need to consider the continuous density function case.

$$
\begin{aligned}
A &= \mathbb{P}(\sum_{i=1}^{S} \pi_i^{1,1} F_i^{1,1}(t_{(k^{1,1})}^{1,1}) - \sum_{i=1}^{S} \pi_i^{1,0} F_i^{1,0}(t_{(k^{1,0})}^{1,0}) > \alpha) \\
&\leq \mathbb{P}(\sum_{i=1}^{S} \pi_i^{1,1} F_i^{1,1}(t_{i,(k_i^{1,1}+1)}^{1,1}) - \sum_{i=1}^{S} \pi_i^{1,0} F_i^{1,0}(t_{i,(k_i^{1,0})}^{1,0}) > \alpha)
\end{aligned}
$$

Considering lemma A.1 and similar result for B, we complete the proof. $\qquad \square$

For the proof of Proposition 3.4, we can adjust the estimation of A by introducing the error generated in rank calculation. Specifically, we show that

*Sketch proof of Proposition 3.4.*

$$A = \mathbb{P}\Big(\sum_{i=1}^{S} \pi_i^{1,1} F_i^{1,1}(t_{(k^{1,1})}^{1,1}) - \sum_{i=1}^{S} \pi_i^{1,0} F_i^{1,0}(t_{(k^{1,0})}^{1,0}) > \alpha\Big)$$

$$\leq \mathbb{P}\Big(\sum_{i=1}^{S} \pi_i^{1,1} F_i^{1,1}(t_{i,\left(k_i^{1,1}+\lfloor \varepsilon n_i^{1,1}\rfloor\right)}^{1,1}) - \sum_{i=1}^{S} \pi_i^{1,0} F_i^{1,0}(t_{i,\left(k_i^{1,0}-\lfloor \varepsilon n_i^{1,0}\rfloor\right)}^{1,0}) > \alpha\Big)$$

$\square$

## A.2   Proof for Proposition 3.5

*Proof for Proposition 3.5.* Note the classifier is

$$\phi = \begin{cases} \mathbb{1}\left\{f(x,0) > \hat{t}_{(k^{1,0})}^{1,0}\right\}, a = 0 \\ \mathbb{1}\left\{f(x,1) > \hat{t}_{(k^{1,1})}^{1,1}\right\}, a = 1 \end{cases}$$

So we can calculate the mis-classification error:

$$\begin{aligned}
\mathbb{P}(Y \neq \hat{Y}) &= \mathbb{P}(Y = 1, \hat{Y} = 0) + \mathbb{P}(Y = 0, \hat{Y} = 1) \\
&= \mathbb{P}(Y = 1, \hat{Y} = 0, A = 0) + \mathbb{P}(Y = 1, \hat{Y} = 0, A = 1) \\
&\quad + \mathbb{P}(Y = 0, \hat{Y} = 1, A = 0) + \mathbb{P}(Y = 0, \hat{Y} = 1, A = 1) \\
&= \sum_{i=1}^{S} \pi_i \big[\mathbb{P}_i(Y = 1, \hat{Y} = 0, A = 0) + \mathbb{P}_i(Y = 1, \hat{Y} = 0, A = 1) \\
&\quad + \mathbb{P}_i(Y = 0, \hat{Y} = 1, A = 0) + \mathbb{P}_i(Y = 0, \hat{Y} = 1, A = 1)\big]
\end{aligned} \tag{22}$$

For ecah specific i, we have

$$\begin{aligned}
&\mathbb{P}_i(Y = 1, \hat{Y} = 0, A = 0) \\
&= \mathbb{P}_i(\hat{Y} = 0 \mid Y = 1, A = 0)\mathbb{P}_i(Y = 1, A = 0) \\
&= \mathbb{E}\left[\mathbb{P}_i(f(x,0) \leq \hat{t}_{(k^{1,0})}^{1,0} \mid Y = 1, A = 0) \mid \hat{t}_{(k^{1,0})}^{1,0}\right] p_0^i p_{Y,0}^i \\
&\leq \mathbb{E}\left[\mathbb{P}_i(f(x,0) \leq t_{i,\left(\hat{k}_i^{1,0}+\lfloor \varepsilon n_i^{1,0}\rfloor+1\right)}^{1,0} \mid Y = 1, A = 0) \mid t_{i,\left(\hat{k}_i^{1,0}+\lfloor \varepsilon n_i^{1,0}\rfloor+1\right)}^{1,0}\right] p_0^i p_{Y,0}^i \\
&= \mathbb{E}\left[F_i^{1,0}(t_{i,\left(\hat{k}_i^{1,0}+\lfloor \varepsilon n_i^{1,0}\rfloor+1\right)}^{1,0}) \mid t_{i,\left(\hat{k}_i^{1,0}+\lfloor \varepsilon n_i^{1,0}\rfloor+1\right)}^{1,0}\right] p_0^i p_{Y,0}^i \\
&= \frac{\hat{k}_i^{1,0} + \lfloor \varepsilon n_i^{1,0}\rfloor + 1}{n_i^{1,0} + 1} p_0^i p_{Y,0}^i
\end{aligned}$$

By the similar reasoning, we point out that

$$\mathbb{P}_i(Y = 1, \hat{Y} = 0, A = 0) \geq \frac{\hat{k}_i^{1,0} - \lfloor \varepsilon n_i^{1,0}\rfloor}{n_i^{1,0} + 1} p_0^i p_{Y,0}^i$$

and thus we have

$$\left|\mathbb{P}_i(Y = 1, \hat{Y} = 0, A = 0) - \frac{\hat{k}_i^{1,0} + 0.5}{n_i^{1,0} + 1} p_0^i p_{Y,0}^i\right| \leq \frac{\lfloor \varepsilon n_i^{1,0}\rfloor + 0.5}{n_i^{1,0} + 1} p_0^i p_{Y,0}^i \tag{23}$$

Moreover, we have

$$
\begin{aligned}
&\mathbb{P}_i(Y = 0, \hat{Y} = 1, A = 0) \\
&= \mathbb{P}_i(\hat{Y} = 1 \mid Y = 0, A = 0)\mathbb{P}_i(Y = 0, A = 0) \\
&= \mathbb{E}\left[\mathbb{P}_i\left(f(x,0) \geq \hat{t}_{(k^{0,0})}^{0,0} \mid Y = 0, A = 0\right) \mid \hat{t}_{(k^{0,0})}^{0,0}\right] p_0^i(1 - p_{Y,0}^i) \\
&\geq \mathbb{E}\left[\mathbb{P}_i\left(f(x,0) \geq t_{i,(\hat{k}_i^{0,0}+\lfloor \varepsilon n_i^{0,0}\rfloor+1)}^{0,0} \mid Y = 0, A = 0\right) \mid t_{i,(\hat{k}_i^{0,0}+\lfloor \varepsilon n_i^{0,0}\rfloor+1)}^{0,0}\right] p_0^i(1 - p_{Y,0}^i) \\
&= \mathbb{E}\left[1 - F_i^{0,0}(t_{i,(\hat{k}_i^{0,0}+\lfloor \varepsilon n_i^{0,0}\rfloor+1)}^{0,0}) \mid t_{i,(\hat{k}_i^{0,0}+\lfloor \varepsilon n_i^{0,0}\rfloor+1)}^{0,0}\right] p_0^i q_{Y,0}^i \\
&= \frac{n_i^{0,0} - \hat{k}_i^{0,0} - \lfloor \varepsilon n_i^{0,0}\rfloor}{n_i^{0,0} + 1} p_0^i(1 - p_{Y,0}^i)
\end{aligned}
$$

Similar, we have

$$
\mathbb{P}_i(Y = 0, \hat{Y} = 1, A = 0) \leq \frac{n_i^{0,0} - \hat{k}_i^{0,0} + \lfloor \varepsilon n_i^{0,0}\rfloor + 1}{n_i^{0,0} + 1} p_0^i(1 - p_{Y,0}^i),
$$

and combining these two result, we get

$$
\left|\mathbb{P}_i(Y = 0, \hat{Y} = 1, A = 0) - \frac{n_i^{0,0} - \hat{k}_i^{0,0} + 0.5}{n_i^{0,0} + 1} p_0^i q_{Y,0}^i\right| \leq \frac{\lfloor \varepsilon n_i^{0,0}\rfloor + 0.5}{n_i^{0,0} + 1} p_0^i(1 - p_{Y,0}^i) \tag{24}
$$

Following similar process of inequality 23 and 24, we can also show that

$$
\left|\mathbb{P}_i(Y = 1, \hat{Y} = 0, A = 1) - \frac{\hat{k}_i^{1,1} + 0.5}{n_i^{1,1} + 1} p_1^i p_{Y,1}^i\right| \leq \frac{\lfloor \varepsilon n_i^{1,1}\rfloor + 0.5}{n_i^{1,1} + 1} p_1^i p_{Y,1}^i \tag{25}
$$

$$
\left|\mathbb{P}_i(Y = 0, \hat{Y} = 1, A = 1) - \frac{n_i^{0,1} - \hat{k}_i^{0,1} + 0.5}{n_i^{0,1} + 1} p_1^i(1 - p_{Y,1}^i)\right| \leq \frac{\lfloor \varepsilon n_i^{0,1}\rfloor + 0.5}{n_i^{0,1} + 1} p_1^i(1 - p_{Y,1}^i) \tag{26}
$$

Combining Inequality 23-26 into equation 22, we complete our proof. □

## A.3   Proof of Proposition 3.6

*Proof of Proposition 3.6.* Note the output classifier is

$$
\phi = \begin{cases}
\mathbb{1}\left\{f(x,0) > \hat{t}_{(k^{1,0})}^{1,0}\right\}, a = 0 \\
\mathbb{1}\left\{f(x,1) > \hat{t}_{(k^{1,1})}^{1,1}\right\}, a = 1
\end{cases}
$$

we have:

$$\mathbb{P}(|DEO(\phi)| \preceq (\alpha, \alpha))$$

$$\geq 1 - \mathbb{P}\left(\left|F^{1,1}\left(t^{1,1}_{(k^{1,1})}\right) - F^{1,0}\left(t^{1,0}_{(k^{1,0})}\right)\right| > \alpha\right) - \mathbb{P}\left(\left|F^{0,1}\left(t^{1,1}_{(k^{1,1})}\right) - F^{0,0}\left(t^{1,0}_{(k^{1,0})}\right)\right| > \alpha\right)$$

$$= 1 - \mathbb{P}\left(\sum_{i=1}^{S} \pi_i^{1,1} F_i^{1,1}\left(t^{1,1}_{(k^{1,1})}\right) - \sum_{i=1}^{S} \pi_i^{1,0} F_i^{1,0}\left(t^{1,0}_{(k^{1,0})}\right) > \alpha\right)$$

$$- \mathbb{P}\left(\sum_{i=1}^{S} \pi_i^{1,1} F_i^{1,1}\left(t^{1,1}_{(k^{1,1})}\right) - \sum_{i=1}^{S} \pi_i^{1,0} F_i^{1,0}\left(t^{1,0}_{(k^{1,0})}\right) < -\alpha\right)$$

$$- \mathbb{P}\left(\sum_{i=1}^{S} \pi_i^{0,1} F_i^{0,1}\left(t^{1,1}_{(k^{1,1})}\right) - \sum_{i=1}^{S} \pi_i^{0,0} F_i^{0,0}\left(t^{1,0}_{(k^{1,0})}\right) > \alpha\right)$$

$$- \mathbb{P}\left(\sum_{i=1}^{S} \pi_i^{0,1} F_i^{0,1}\left(t^{1,1}_{(k^{1,1})}\right) - \sum_{i=1}^{S} \pi_i^{0,0} F_i^{0,0}\left(t^{1,0}_{(k^{1,0})}\right) < -\alpha\right)$$

The remainder of the proof is similar to the proof for Proposition 3.2

$\square$

Building upon Proposition 3.6, we can further prove Theorem 4.3 using a similar approach as in Theorem 4.2.

## A.4   Proof for Theorem 4.2

To begin with, the Fair Bayes-optimal Classifiers under Equality of Opportunity is defined by following lemma, wherein $\eta_a(x) := \mathbb{P}(Y = 1 \mid A = a, X = x)$ stands for the proportion of group $Y = 1$ conditioned on $A$ and $X$.

**Lemma A.2** (Theorem E.4 in (Zeng et al., 2022)). *Let $E^\star = \mathrm{DEOO}(f^\star)$. For any $\alpha > 0$, all fair Bayes-optimal classifiers $f^\star_{E,\alpha}$ under the fairness constraint $|\mathrm{DEOO}(f)| \leq \alpha$ are given as follows:*
*- When $|E^\star| \leq \alpha, f^\star_{E,\alpha} = f^\star$*
*- When $|E^\star| > \alpha$, suppose $\mathbb{P}_{X|A=1,Y=1}(\eta_1(X) = \frac{p_1 p_{Y,1}}{2(p_1 p_{Y,1} - t^\star_{E,\alpha})}) = 0$, then for all $x \in \mathcal{X}$ and $a \in \mathcal{A}$,*

$$f^\star_{E,\alpha}(x, a) = I(\eta_a(x) > \frac{p_a p_{Y,a}}{2 p_a p_{Y,a} + (1 - 2a) t^\star_{E,\alpha}})$$

*where $t^\star_{E,\alpha}$ is defined as*

$$t^\star_{E,\alpha} = \sup\left\{ t : \mathbb{P}_{Y|A=1,Y=1}\left(\eta_1(X) > \frac{p_1 p_{Y,1}}{2 p_1 p_{Y,1} - t}\right) \right.$$
$$\left. > \mathbb{P}_{Y|A=0,Y=1}\left(\eta_0(X) > \frac{p_0 p_{Y,0}}{2 p_0 p_{Y,0} + t}\right) + \frac{E^\star}{|E^\star|}\alpha \right\}.$$

**Lemma A.3** (Hoeffding's inequality). *Let $X_1, \ldots, X_n$ be independent random variables. Assume that $X_i \in [m_i, M_i]$ for every $i$. Then, for any $t > 0$, we have*

$$\mathbb{P}\left\{ \sum_{i=1}^{n} (X_i - \mathbb{E}X_i) \geq t \right\} \leq e^{-\frac{2t^2}{\sum_{i=1}^{n}(M_i - m_i)^2}}$$

Then, we introduce several lemma to prove Theorem 4.2.

**Lemma A.4.** *For a distribution $F$ with a continuous density function, suppose $q(x)$ denotes the quantile of $x$ under $F$, then for $x > y$, we have $F_{(-)}(x-y) \leq q(x) - q(y) \leq F_{(+)}(x-y)$, where $F_{(-)}(x)$ and $F_{(+)}(x)$ are two monotonically increasing functions, $F_{(-)}(\epsilon) > 0, F_{(+)}(\epsilon) > 0$ for any $\epsilon > 0$ and $\lim_{\epsilon \to 0} F_{(-)}(\epsilon) = \lim_{\epsilon \to 0} F_{(+)}(\epsilon) = 0$.*

*Proof of Lemma A.4.* Since the domain of $q(x)$ is a closed set and $q(x)$ is continuous, we know that $q(x)$ is uniformly continuous. Thus we can easily find $F_{(+)}$ to satisfy the RHS. For $F_{(-)}$, we simply define $F_{(-)}(t) = \inf_x \{q(x+t) - q(t)\}$. Since $q(x+t) - q(t) > 0$ for $t > 0$ and the domain of $x$ is a closed set, we have $F_{(-)}(\epsilon) > 0$ for $\epsilon > 0$ and $\lim_{\epsilon \to 0} F_{(-)}(\epsilon) = 0$. Now we complete the proof. □

*Proof for theorem 4.2.* In fact, (1) of the theorem is a direct application of Proposition 3.4, so we only need to prove (2). In partcular, the main idea of our proof is to find a bridge between fair Bayes optimal classifier and our output classifier.

To begin with, we show that there exist a classifier in our set which is quite similar with fair Bayes optimal classifier. Suppose the fair Bayes optimal classifier has the form $\phi^*_{\alpha'}(x,a) = \mathbb{I}\{f^*(x,a) > \lambda^*_a\}$ and our output classifier is of the form $\hat{\phi}(x,a) = \mathbb{1}\{f(x,a) > \lambda_a\}$.

For any $\epsilon > 0$, by Lemma A.4, we know that above than a positive probability $F^{1,a}_{i,(-)}(2\epsilon)$, $t^{1,a}_i$ would fall in the interval $[\lambda^*_a - \epsilon, \lambda^*_a + \epsilon]$ for each client $i$. By the definition of $\varepsilon$-approximate quantile, we have at most with probability $\prod_{i=1}^S \left(1 - F^{1,0}_{i,(-)}(2\epsilon)\right)^{n^{1,0}_i} + \prod_{i=1}^S \left(1 - F^{1,1}_{i,(-)}(2\epsilon)\right)^{n^{1,1}_i}$, there exists $a \in \{0,1\}$ such that all $t^{1,a}_{i,(k)}$ fall out of $[\lambda^*_a - \epsilon, \lambda^*_a + \epsilon]$. Thus, with probability $1 - \prod_{i=1}^S \left(1 - F^{1,0}_{i(-)}(2\epsilon)\right)^{n^{1,0}_i} - \prod_{i=1}^S \left(1 - F^{1,1}_{i(-)}(2\epsilon)\right)^{n^{1,1}_i}$, for $a \in \{0,1\}$, there would exist i such that there exists at least one $t^{1,a}_i$ in $[\lambda^*_a - \epsilon, \lambda^*_a + \epsilon]$. So with $1 - \prod_{i=1}^S \left(1 - F^{1,0}_{i(-)}(2\epsilon)\right)^{n^{1,0}_i} - \prod_{i=1}^S \left(1 - F^{1,1}_{i(-)}(2\epsilon)\right)^{n^{1,1}_i} - \delta(n^{1,0}) - \delta(n^{1,1})$, there exist a classifier $\phi_0(x,a) = \mathbb{1}\left\{f(x,a) > \hat{t}^{1,a}_*\right\}$ such that $\hat{t}^{1,a}_* \in [\lambda^*_a - \epsilon - \gamma\varepsilon, \lambda^*_a + \epsilon + \gamma\varepsilon]$. We also denote $\phi^*_0(x,a) = \mathbb{1}\left\{f^*(x,a) > t^{1,a}_*\right\}$. Given the threshold is quite close, we further prove that the accuracy is quite close with a high probability. Actually, we have

$$
\begin{aligned}
&|\mathbb{P}\left(\phi_0(x,a) \neq Y\right) - \mathbb{P}\left(\phi^*_{\alpha'}(x,a) \neq Y\right)| \\
\leq& |\mathbb{P}\left(\phi_0(x,a) \neq Y\right) - \mathbb{P}\left(\phi^*_0(x,a) \neq Y\right)| + |\mathbb{P}\left(\phi^*_0(x,a) \neq Y\right) - \mathbb{P}\left(\phi^*_{\alpha'}(x,a) \neq Y\right)| \\
\leq& \mathbb{P}\left(t^{1,a}_* - \epsilon_0 \leq f^*(x,a) \leq t^{1,a}_* + \epsilon_0\right) + \mathbb{P}\left(\min\left\{t^{1,a}_*, \lambda^*_a\right\} \leq f^*(x,a) \leq \max\left\{t^{1,a}_*, \lambda^*_a\right\}\right) \quad (27) \\
\leq& F^*_{(+)}(2\epsilon_0) + F^*_{(+)}\left(\max\left\{t^{1,a}_*, \lambda^*_a\right\} - \min\left\{t^{1,a}_*, \lambda^*_a\right\}\right) \\
\leq& F^*_{(+)}(2\epsilon_0) + 2F^*_{(+)}(\epsilon + \gamma\varepsilon)
\end{aligned}
$$

with probability $1 - \prod_{i=1}^S \left(1 - F^{1,0}_{i,(-)}(2\epsilon)\right)^{n^{1,0}_i} - \prod_{i=1}^S \left(1 - F^{1,1}_{i,(-)}(2\epsilon)\right)^{n^{1,1}_i} - \delta(n^{1,0}) - \delta(n^{1,1})$. Further we point out that

$$|| \text{DEOO}(\phi_0)| - |\text{DEOO}(\phi_{\alpha'}^*)||$$
$$\leq ||\text{DEOO}(\phi_0)| - |\text{DEOO}(\phi_0^*)| + |DEOO(\phi_0^*)| - |\text{DEOO}(\phi_{\alpha'}^*)||$$
$$= ||\mathbb{P}(f > t_*^{1,0} \mid Y = 1, A = 0) - \mathbb{P}(f > t_*^{1,1} \mid Y = 1, A = 1)|$$
$$- |\mathbb{P}\left(f^* > t_*^{1,0} \mid Y = 1, A = 0\right) - \mathbb{P}\left(f^* > t_*^{1,1} \mid Y = 1, A = 1\right)||$$
$$+ ||\mathbb{P}\left(f^* > t_*^{1,0} \mid Y = 1, A = 0\right) - \mathbb{P}\left(f^* > t_*^{1,1} \mid Y = 1, A = 1\right)|$$
$$- |\mathbb{P}(f^* > \lambda_0^* \mid Y = 1, A = 0) - \mathbb{P}(f^* > \lambda_1^* \mid Y = 1, A = 1)||$$
$$\leq |\mathbb{P}\left(f > t_*^{1,0} \mid Y = 1, A = 0\right) - \mathbb{P}\left(f^* > t_*^{1,0} \mid Y = 1, A = 0\right)|$$
$$+ |\mathbb{P}\left(f > t_*^{1,1} \mid Y = 1, A = 1\right) - \mathbb{P}\left(f^* > t_*^{1,1} \mid Y = 1, A = 1\right)|$$
$$+ ||\mathbb{P}\left(f^* > t_*^{1,0} \mid Y = 1, A = 0\right) - \mathbb{P}\left(f^* > t_*^{1,1} \mid Y = 1, A = 1\right)|$$
$$- |\mathbb{P}(f^* > \lambda_0^* \mid Y = 1, A = 0) - \mathbb{P}(f^* > \lambda_1^* \mid Y = 1, A = 1)||$$
$$\leq \mathbb{P}\left(t_*^{1,0} - \epsilon_0 \leq f^*(x, a) \leq t_*^{1,0} + \epsilon_0\right) + \mathbb{P}\left(t_*^{1,1} - \epsilon_0 \leq f^*(x, a) \leq t_*^{1,1} + \epsilon_0\right)$$
$$+ |\mathbb{P}\left(f^* > t_*^{1,0} \mid Y = 1, A = 0\right) - \mathbb{P}\left(f^* > t_*^{1,1} \mid Y = 1, A = 1\right)$$
$$- \mathbb{P}(f^* > \lambda_0^* \mid Y = 1, A = 0) + \mathbb{P}(f^* > \lambda_1^* \mid Y = 1, A = 1)|$$
$$\leq 2F_{(+)}^*(2\epsilon_0) + \mathbb{P}\left(\min\left\{t_*^{1,a}, \lambda_a^*\right\} \leq f^*(x, a) \leq \max\left\{t_*^{1,a}, \lambda_a^*\right\}\right)$$
$$\leq 2F_{(+)}^*(2\epsilon_0) + F_{(+)}^*\left(\max\left\{t_*^{1,a}, \lambda_a^*\right\} - \min\left\{t_*^{1,a}, \lambda_a^*\right\}\right)$$
$$\leq 2F_{(+)}^*(2\epsilon_0) + 2F_{(+)}^*(\epsilon + \gamma\varepsilon)$$

Thus, we know that

$$|\text{DEOO}(\phi_0)| \leq |DEOO(\phi_{\alpha'}^*)| + 2F_{(+)}^*(2\epsilon_0) + 2F_{(+)}^*(\epsilon + \gamma\varepsilon) = \alpha' + 2F_{(+)}^*(2\epsilon_0) + 2F_{(+)}^*(\epsilon + \gamma\varepsilon)$$

If $F_{(+)}^*(\epsilon + \gamma\varepsilon) \leq \frac{\alpha - \alpha'}{2} - F_{(+)}^*(2\epsilon_0)$, then there will exist at least one feasible classifier in the candidate set.

On the other hand, we could prove that the output classifier is quite similar with $\phi_0$ we mentioned above.

By Proposition 3.5, for any $\phi \in K$, $\hat{q}_{Y,a}^i = 1 - \hat{p}_{Y,a}^i$, we have

$$\left| \mathbb{P}\left(\phi(x, a) \neq Y\right) - \sum_{i=1}^S \pi_i \left[ \frac{\hat{k}_i^{1,0} + 0.5}{n_i^{1,0} + 1} p_0^i p_{Y,0}^i + \frac{\hat{k}_i^{1,1} + 0.5}{n_i^{1,1} + 1} p_1^i p_{Y,1}^i \right.\right.$$
$$\left.\left. + \frac{n_i^{0,0} + 0.5 - \hat{k}_i^{0,0}}{n_i^{0,0} + 1} p_0^i q_{Y,0}^i + \frac{n_i^{0,1} + 0.5 - \hat{k}_i^{0,1}}{n_i^{0,1} + 1} p_1^i q_{Y,1}^i \right] \right| \leq \theta \tag{28}$$

Therefore, we only need to check the influence induced by using $\hat{p}_a^i$ and $\hat{p}_{Y,a}^i$, instead of $p_0^i$ and $p_{Y,0}^i$. In detail, we point out this influence can be estimated by Hoeffding's inequality as follow:

Since $\hat{p}_a^i = \frac{n_i^{1,a} + n_i^{0,a}}{n_i}$ and $\hat{p}_{Y,a}^i = \frac{n_i^{1,a}}{n_i^{0,a} + n_i^{1,a}}$, we have $\frac{n_i^{1,a} + n_i^{0,a}}{n_i} = \frac{\sum_{j=1}^{n_i} \mathbb{1}\{Z_j^a = 1\}}{n}$ and $\frac{n_i^{1,a}}{n_i^{0,a} + n_i^{1,a}} = \frac{\sum_{j=1}^{n_i^{0,a} + n_i^{1,a}} \mathbb{1}\{Z_j^{Y,a} = 1\}}{n_i^{0,a} + n_i^{1,a}}$, where $Z_j^a \sim B\left(1, p_a^i\right)$ and $Z_j^{Y,a} \sim B\left(1, p_{Y,a}^i\right)$.

Thus, from Hoeffding's inequality, we have

$$\mathbb{P}\left(\left|\hat{p}_a^i - p_a^i\right| \geq \sqrt{\frac{n_i^{0,a}}{n_i}}\epsilon\right) \leq 2e^{-2n_i^{0,a}\epsilon^2}$$

For the same reason, we have we have

$$\mathbb{P}\left(\left|\hat{p}_{Y,a}^i - p_{Y,a}^i\right| \geq \sqrt{\frac{n_i^{0,a}}{n_i}}\epsilon\right) \leq 2e^{-2n_i^{0,a}\epsilon^2}$$

So, we have with probability $1 - 4\sum_{i=1}^{S} e^{-2n_i^{0,a}\epsilon^2}$

$$\begin{cases} \left|\hat{p}_a^i - p_a^i\right| \leq \sqrt{\frac{n_i^{0,a}}{n_i}}\epsilon \\ \left|\hat{p}_{Y,a}^i - p_{Y,a}^i\right| \leq \sqrt{\frac{n_i^{0,a}}{n_i^{*,a}}}\epsilon \end{cases},$$

where $n_i^{*,a} = (n_i^{0,a} + n_i^{1,a})$.

Thus, with probability $1 - 4\sum_{a=0}^{1}\sum_{i=1}^{S} e^{-2n_i^{0,a}\epsilon^2}$,

$$\left|\mathbb{P}\left(\hat{\phi}_i(x,a) \neq Y\right) - \hat{\mathbb{P}}\left(\hat{\phi}_i(x,a) \neq Y\right)\right|$$

$$\leq \left|\sum_{i=1}^{S}\pi^i\left[\frac{\hat{k}_i^{1,0}+0.5}{n_i^{1,0}+1}p_0^i p_{Y,0}^i + \frac{\hat{k}_i^{1,1}+0.5}{n_i^{1,1}+1}p_1^i p_{Y,1}^i + \frac{n_i^{0,0}+0.5-\hat{k}_i^{0,0}}{n_i^{0,0}+1}p_0^i q_{Y,0}^i + \frac{n_i^{0,1}+0.5-\hat{k}_i^{0,1}}{n_i^{0,1}+1}p_1^i q_{Y,1}^i\right]\right.$$

$$\left. - \sum_{i=1}^{S}\pi^i\left[\frac{\hat{k}_i^{1,0}+0.5}{n_i^{1,0}+1}\hat{p}_0^i \hat{p}_{Y,0}^i + \frac{\hat{k}_i^{1,1}+0.5}{n_i^{1,1}+1}\hat{p}_1^i \hat{p}_{Y,1}^i + \frac{n_i^{0,0}+0.5-\hat{k}_i^{0,0}}{n_i^{0,0}+1}\hat{p}_0^i \hat{q}_{Y,0}^i + \frac{n_i^{0,1}+0.5-\hat{k}_i^{0,1}}{n_i^{0,1}+1}\hat{p}_1^i \hat{q}_{Y,1}^i\right]\right|$$

$$+ \sum_{i=1}^{S}\pi_i\left[e_i^{0,0}p_0^i q_{Y,0}^i + e_i^{0,1}p_0^i p_{Y,0}^i + e_i^{1,0}p_1^i q_{Y,1}^i + e_i^{1,1}p_1^i p_{Y,1}^i\right]$$

$$= \sum_{i=1}^{S}\pi_i\left[e_i^{0,0}p_0^i q_{Y,0}^i + e_i^{0,1}p_0^i p_{Y,0}^i + e_i^{1,0}p_1^i q_{Y,1}^i + e_i^{1,1}p_1^i p_{Y,1}^i\right] + \left|\sum_{i=1}^{S}\pi^i(A_i - \hat{A}_i)\right|$$

$$\tag{29}$$

For $A_i - \hat{A}_i$, we have

$$A_i - \hat{A}_i \leq \epsilon \left[ \sqrt{\frac{n_i^{0,0}}{n_i^{*,0}}} \frac{\hat{k}_i^{1,0} + 0.5}{n^{1,0} + 1} \left( p_0^i + p_{Y,0}^i \right) + \sqrt{\frac{n_i^{0,1}}{n_i^{*,1}}} \frac{\hat{k}_i^{1,1} + 0.5}{n^{1,1} + 1} \left( p_1^i + p_{Y,1}^i \right) \right]$$

$$+ \epsilon^2 \left( \frac{n_i^{0,0}}{n_i^{*,0}} \frac{\hat{k}_i^{1,0} + 0.5}{n^{1,0} + 1} + \frac{n_i^{0,1}}{n_i^{*,1}} \frac{\hat{k}_i^{1,1} + 0.5}{n^{1,1} + 1} \right) + \frac{n^{0,0} + 0.5 - \hat{k}_i^{0,0}}{n^{0,0} + 1} \sqrt{\frac{n_i^{0,0}}{n_i^{*,0}}} \epsilon \left[ \sqrt{\frac{n_i^{0,0}}{n_i^{*,0}}} \epsilon + p_0^i + p_{Y,0}^i + 1 \right]$$

$$+ \frac{n^{0,1} + 0.5 - \hat{k}_i^{0,1}}{n^{0,1} + 1} \sqrt{\frac{n_i^{0,1}}{n_i^{*,1}}} \epsilon \left[ \sqrt{\frac{n_i^{0,1}}{n_i^{*,1}}} \epsilon + p_1^i + p_{Y,1}^i + 1 \right]$$

$$\leq \epsilon \left[ \sqrt{\frac{n_i^{0,0}}{n_i^{*,0}}} \left( p_0^i + p_{Y,0}^i \right) + \sqrt{\frac{n_i^{0,1}}{n_i^{*,1}}} \left( p_1^i + p_{Y,1}^i \right) \right] + \epsilon^2 \left( \frac{n_i^{0,0}}{n_i^{*,0}} + \frac{n_i^{0,1}}{n_i^{*,1}} \right)$$

$$+ \sqrt{\frac{n_i^{0,0}}{n_i^{*,0}}} \epsilon \left[ \sqrt{\frac{n_i^{0,0}}{n_i^{*,0}}} \epsilon + p_0^i + p_{Y,0}^i + 1 \right] + \sqrt{\frac{n_i^{0,1}}{n_i^{*,1}}} \epsilon \left[ \sqrt{\frac{n_i^{0,1}}{n_i^{*,1}}} \epsilon + p_1^i + p_{Y,1}^i + 1 \right]$$

$$\leq 4\epsilon + 2\epsilon^2 + 2\epsilon^2 + 6\epsilon$$

$$= 4\epsilon^2 + 10\epsilon$$

(30)

Combining Inequality , we complete the proof. $\square$

## A.5 Proof of the Results for Label Shift Case

*Proof for Proposition 6.2.* Note the classifier is

$$\phi = \begin{cases} \mathbb{1}\left\{ f(x,0) > \hat{t}_{(k^{1,0})}^{1,0} \right\}, a = 0 \\ \mathbb{1}\left\{ f(x,1) > \hat{t}_{(k^{1,1})}^{1,1} \right\}, a = 1 \end{cases}$$

So we can calculate the mis-classification error in $P_{S+1}$. Denoted $\mathbb{P}_{S+1}$ the probability measure under the $P_{S+1}$ distribution, we have:

$$
\begin{aligned}
\mathbb{P}_{S+1}(Y \neq \hat{Y}) &= \mathbb{P}_{S+1}(Y = 1, \hat{Y} = 0) + \mathbb{P}_{S+1}(Y = 0, \hat{Y} = 1) \\
&= \mathbb{P}_{S+1}(Y = 1, \hat{Y} = 0, A = 0) + \mathbb{P}_{S+1}(Y = 1, \hat{Y} = 0, A = 1) \\
&+ \mathbb{P}_{S+1}(Y = 0, \hat{Y} = 1, A = 0) + \mathbb{P}_{S+1}(Y = 0, \hat{Y} = 1, A = 1) \\
&= \mathbb{P}(Y = 1, \hat{Y} = 0, A = 0 \mid (X,Y,A) \sim P_{S+1}) + \mathbb{P}(Y = 1, \hat{Y} = 0, A = 1 \mid (X,Y,A) \sim P_{S+1}) \\
&+ \mathbb{P}(Y = 0, \hat{Y} = 1, A = 0 \mid (X,Y,A) \sim P_{S+1}) + \mathbb{P}(Y = 0, \hat{Y} = 1, A = 1 \mid (X,Y,A) \sim P_{S+1}) \\
&= \mathbb{P}(\hat{Y} = 0 \mid Y = 1, A = 0)p_0^{S+1}p_{Y,0}^{S+1} + \mathbb{P}(\hat{Y} = 0 \mid Y = 1, A = 1)p_1^{S+1}p_{Y,1}^{S+1} \\
&+ \mathbb{P}(\hat{Y} = 1 \mid Y = 0, A = 0)p_0^{S+1}(1 - p_{Y,0}^{S+1}) + \mathbb{P}(\hat{Y} = 1 \mid Y = 0, A = 1)p_1^{S+1}(1 - p_{Y,1}^{S+1}) \\
&= \sum_{i=1}^{S} \pi_i^{1,0}\mathbb{P}_i(\hat{Y} = 0 \mid Y = 1, A = 0)p_0^{S+1}p_{Y,0}^{S+1} + \sum_{i=1}^{S} \pi_i^{1,1}\mathbb{P}(\hat{Y} = 0 \mid Y = 1, A = 1)p_1^{S+1}p_{Y,1}^{S+1} \\
&+ \sum_{i=1}^{S} \pi_i^{0,0}\mathbb{P}(\hat{Y} = 1 \mid Y = 0, A = 0)p_0^{S+1}(1 - p_{Y,0}^{S+1}) \\
&+ \sum_{i=1}^{S} \pi_i^{0,1}\mathbb{P}(\hat{Y} = 1 \mid Y = 0, A = 1)p_1^{S+1}(1 - p_{Y,1}^{S+1}) \\
&= \sum_{i=1}^{S} \pi_i \big[ w^{0,0}\mathbb{P}_i(Y = 1, \hat{Y} = 0, A = 0) + w^{0,1}\mathbb{P}_i(Y = 1, \hat{Y} = 0, A = 1) \\
&+ w^{1,0}\mathbb{P}_i(Y = 0, \hat{Y} = 1, A = 0) + w^{1,1}\mathbb{P}_i(Y = 0, \hat{Y} = 1, A = 1) \big]
\end{aligned}
\tag{31}
$$

Then, since estimating $\mathbb{P}_i(Y = 0, \hat{Y} = y, A = a)$ shares similarities with the approach outlined in Proposition 3.5. This similarity in the estimation process allows us to successfully complete our proof. $\qquad\square$

Given proof for Proposition 6.2, proof for Theorem 6.3 is similar to Proof for Theorem 4.2

## A.6 Extension to Multi-Label Classification Case

**Definition A.5.** (Equality of Opportunity, Multiple labels(Liu et al., 2023)) A classifier satisfies Equality of Opportunity if it satisfies :

$$
\hat{\boldsymbol{Y}} \perp A \mid \boldsymbol{Y} = \boldsymbol{y}_{adv},
$$

where $\boldsymbol{Y} \in \{0,1\}^m$ and $\boldsymbol{y}_{adv}$ denotes some advantaged label where only favorable outcomes.

**Definition A.6.** (Multi-label Score-based Classifier) A Multi-label score-based classifier is an element-wise indication function, where the j-th component of $\hat{\boldsymbol{Y}}$ satisfies $\hat{Y}_j = \phi_j(x,a) = \mathbb{1}\{f_j(x,a) > c_j\}$ for a measurable score function $f : \mathcal{X} \times \{0,1\} \to [0,1]$ and a constant threshold $c_j > 0$.

Considering relaxing the aforementioned Equality of Opportunity constraint, we introduce a fairness indicator as follow:

$$
DEOOM_{\boldsymbol{y}}(\phi) = \big| \mathbb{P}[\hat{\boldsymbol{Y}} = \boldsymbol{y} \mid A = 0, \boldsymbol{Y} = \boldsymbol{Y}_{adv}] - \mathbf{P}[\hat{\boldsymbol{Y}} = \boldsymbol{y} \mid A = 1, \boldsymbol{Y} = \boldsymbol{Y}_{adv}] \big|,
$$

where $\boldsymbol{y}$ can be considered as either certain advantageous labels or as a collection of advantageous labels (at this point, '=' is replaced by '$\in$').

Additionally, we consider an iterative Q-digest approach. At each client, our process involves constructing a Q-digest initially for the first component of the score $f(x)$. Subsequently, at each leaf node, we include a Q-digest for the second component of score $f(x)$ associated with the leaf node's first component. Repeating this procedure iteratively allows us to generate a sketch for the multidimensional score function $f(x)$. Assuming the parameter is appropriately set to achieve an $\varepsilon_j$-approximate quantile and rank for the $j$-th component, we arrive at the following result.

**Proposition A.7.** *Under Assumption 3.1, for $a \in \{0,1\}$, consider*

$$\boldsymbol{q}^{\boldsymbol{y}_{adv},a} = (q_1^{\boldsymbol{y}_{adv},a}, q_2^{\boldsymbol{y}_{adv},a}, ..., q_m^{\boldsymbol{y}_{adv},a}) \in [0,1]^m,$$

$n_{i,(j)}^{\boldsymbol{y}_{adv},a}$ *is the estimation of* $|N_{i,(j)}^{\boldsymbol{y}_{adv},a}|$, $N_{i,(j)}^{\boldsymbol{y}_{adv},a} = \{f_j(x) \mid x \text{ belongs to Client } i, , Y = \boldsymbol{y}_{adv}, A = a, (f_l(x) - t_l)y_l^* \geq 0, l = 1, \cdots, j-1\}|$ *and* $t_j^{\boldsymbol{y}_{adv}}$ *is estimation of* $q_j$ *quantile of* $N_{*,(j)}^{\boldsymbol{y}_{adv},a}$ *(the union of* $N_{i,(j)}^{\boldsymbol{y}_{adv},a}$*), where estimations with subscript* $(j)$ *are* $\varepsilon$*-approximate ranks and quantiles,* $\hat{k}_{i,(j)}^{\boldsymbol{y}_{adv},a}$ *represent the estimation local rank of* $t_j^{\boldsymbol{y}_{adv}}$ *in* $N_{i,(j)}^{\boldsymbol{y}_{adv},a}$*, the score-based classifier* $\phi(x,a) = \mathbb{1}\{f(x,a) > t_j^{\boldsymbol{y}_{adv},a}\}$*. Define*

$$h_{\boldsymbol{y}_{adv},a} = \mathbb{P}\left(\sum_{i=1}^S \pi_i^{\boldsymbol{y}_{adv},a} \prod_{j=1}^m g_j\left(Q\left(u_{i,(j)}^{\boldsymbol{y}_{adv},a}, l_j n_{i,(j)}^{\boldsymbol{y}_{adv},a} + 1 - u_{i,(j)}^{\boldsymbol{y}_{adv},a}\right)\right)\right.$$

$$\left. - \sum_{i=1}^S \pi_i^{\boldsymbol{y}_{adv},1-a} \prod_{j=1}^m g_j\left(Q\left(v_{i,(j)}^{\boldsymbol{y}_{adv},1-a}, (2-l_j) n_{i,(j)}^{\boldsymbol{y}_{adv},1-a} + 1 - v_{i,(j)}^{\boldsymbol{y}_{adv},1-a}\right)\right) \geq \alpha\right),$$

*Then we have:*

$$\mathbb{P}(|DEOOM_{\boldsymbol{y}^*}(\phi)| > \alpha) \leq h_{\boldsymbol{y}_{adv},0} + h_{\boldsymbol{y}_{adv},1}, \tag{32}$$

*where* $\pi_i^{\boldsymbol{y}_{adv},a}$ *is similarly defined as in Proposition 3.2,* $l_j = (1 - 2y_j^*)\varepsilon_{j-1}, g_j(Q) = (1 - 2y_j^*)Q + y_j^*, u_{i,(j)}^{\boldsymbol{y}_{adv},a} = y_j^* m_{i,(j)}^{\boldsymbol{y}_{adv},a} + (1 - y_j^*) M_{i,(j)}^{\boldsymbol{y}_{adv},a}$, $v_{i,(j)}^{\boldsymbol{y}_{adv},a} = y_j^* M_{i,(j)}^{\boldsymbol{y}_{adv},a} + (1 - y_j^*) m_{i,(j)}^{\boldsymbol{y}_{adv},a}$, $M_{i,(j)}^{\boldsymbol{y}_{adv},a} = \min\left(\lceil \hat{k}_{i,(j)}^{\boldsymbol{y}_{adv},a} + \varepsilon_j n_{i,(j)}^{\boldsymbol{y}_{adv},a}\rceil, n_{i,(j)}^{\boldsymbol{y}_{adv},a} + 1\right), m_{i,(j)}^{\boldsymbol{y}_{adv},a} = \max\left(\lceil \hat{k}_{i,(j)}^{\boldsymbol{y}_{adv},a} - \varepsilon_j n_{i,(j)}^{\boldsymbol{y}_{adv},a}\rceil, 0\right)$, *and* $Q(\alpha,\beta)$ *are independent random variables and* $Q(\alpha,\beta) \sim Beta(\alpha,\beta)$*. Especially, we define* $Q(0,\beta) = 0$ *and* $Q(\alpha,0) = 1$ *for* $\alpha,\beta \neq 0$*.*

The proposition above can be proved using Lemma A.1 and conditional probability. It is important to note that $\boldsymbol{y}$ and $\boldsymbol{y}_{adv}$ are not necessarily single labels; they can also represent a set of labels with constraints on specific components where values are restricted to 0 or 1 (for $j$ where $y_j^*$ does not have constraint, $t_j$ is set to 0.5, and it is excluded from the construction of $N$ and calculation of $h$). And similarly, the selection can be conducted by minimizing empirical misclassification error.

Considering a high-dimensional extension of Lemma A.4, we have

**Lemma A.8.** *For a distribution $F$ with a continuous density function, suppose $q(x)$ denotes the probability of $X \preceq x$ where $X$ is a random variable under $F$, then for $y \preceq x$, we have $F_{(-)}(||x-y||_2) \leq q(x) - q(y) \leq F_{(+)}(||x-y||_2)$, where $F_{(-)}(x)$ and $F_{(+)}(x)$ are two monotonically*

*increasing functions, $F_{(-)}(\epsilon) > 0, F_{(+)}(\epsilon) > 0$ for any $\epsilon > 0$ and $\lim_{\epsilon \to 0} F_{(-)}(\epsilon) = \lim_{\epsilon \to 0} F_{(+)}(\epsilon) = 0$.*

Therefore, similarly, we have

**Theorem A.9.** *Under Assumption 3.1 and 4.1, given $\alpha' < \alpha$. Suppose $\hat{\phi}$ is the final output of FedFaiREE, we have:*

*(1) **Fairness guarantee.** $|DEOOM_{\mathbf{y}^*}(\hat{\phi})| < \alpha$ with probability $(1-\delta)^N$, where $N$ is the size of the candidate set.*

*(2) **Accuracy guarantee.** Suppose the density distribution functions of $f^*$ under $A = a, Y = 1$ are continuous. When the input classifier $f$ satisfies $||f(x,a) - f^*(x,a)||_2 \leq \epsilon_0$, for any $\epsilon > 0$ such that $M^*_{(+)}(\epsilon + \gamma\varepsilon) \leq \frac{\alpha - \alpha'}{2m} - M^*_{(+)}(2\epsilon_0)$, we have*

$$\mathbb{P}(\hat{\phi}(x,a) \neq Y) - \mathbb{P}\left(\phi^*_{\alpha'}(x,a) \neq Y\right) \leq 2mM^*_{(+)}(2\epsilon_0) + 2mM^*_{(+)}(\epsilon + \gamma\varepsilon_m) + 2\theta + O(\epsilon) \qquad (33)$$

*with probability $1 - (2^{m+1} + 2)\sum_{a=0}^{1}\sum_{i=1}^{S} e^{-2n_i^{0,a}\epsilon^2} - \prod_{i=1}^{S}\left(1 - F_{i(-)}^{\mathbf{y}_{adv},0}(2\epsilon)\right)^{n_i^{\mathbf{y}_{adv},0}} - \prod_{i=1}^{S}\left(1 - F_{i(-)}^{\mathbf{y}_{adv},1}(2\epsilon)\right)^{n_i^{\mathbf{y}_{adv},1}} - \delta$, where $\delta = \sum_{a=0}^{1}\delta^{\mathbf{y}_{adv},a}(n^{\mathbf{y}_{adv},a})$, $\theta = \sum_{i=1}^{S}\left[\pi_i\sum_{a=0}^{1}\sum_{\mathbf{y}} e_i^{\mathbf{y},a}p_a^i p_{\mathbf{y},a}^i\right]$, $e_i^{\mathbf{y},a} = \frac{2\lfloor\varepsilon_m n_i^{\mathbf{y},a}\rfloor + 1}{2(n_i^{\mathbf{y},a}+1)}$, $M^*_{(+)}$ corresponds to the maximum of $F_{(+)}$ associated with $f_j^*$, and the definition of $F_{(+)}$ and $F_{(-)}$ are shown in Lemma A.8.*

# B  Application to More Fairness Notions

In this section, we delve into the application of FedFaiREE on additional fairness concepts.

## B.1  Definition

To begin with, we introduce the definitions of various fairness concepts.

**Definition B.1.** (Equalized Odds (Hardt et al., 2016)) A classifier satisfies Equalized Odds if it satisfies the following equality: $\mathbb{P}_{X|A=1,Y=1}(\widehat{Y} = 1) = \mathbb{P}_{X|A=0,Y=1}(\widehat{Y} = 1)$ and $\mathbb{P}_{X|A=1,Y=0}(\widehat{Y} = 1) = \mathbb{P}_{X|A=0,Y=0}(\widehat{Y} = 1)$.

**Definition B.2** (Demographic Parity)**.** A classifier satisfies Demographic Parity if its prediction $\widehat{Y}$ is statistically independent of the sensitive attribute $A$:

$$\mathbb{P}(\widehat{Y} = 1 \mid A = 1) = \mathbb{P}(\widehat{Y} = 1 \mid A = 0)$$

**Definition B.3** (Predictive Equality)**.** A classifier satisfies Predictive Equality if it achieves the same TNR (or FPR) among protected groups:

$$\mathbb{P}_{X|A=1,Y=0}(\widehat{Y} = 1) = \mathbb{P}_{X|A=0,Y=0}(\widehat{Y} = 1)$$

**Definition B.4** (Overall Accuracy Equality)**.** A classifier satisfies Overall Accuracy Equality if its mis-classification error is statistically independent of the sensitive attribute $A$:

$$\mathbb{P}(\widehat{Y} \neq Y \mid A = 1) = \mathbb{P}(\widehat{Y} \neq Y \mid A = 0)$$

Similar to *DEOO*, we define the following indicators:

$$\text{DDP} = \mathbb{P}_{X|A=1}(\widehat{Y} = 1) - \mathbb{P}_{X|A=0}(\widehat{Y} = 1) \tag{34}$$

$$\text{DPE} = \mathbb{P}_{X|A=1,Y=0}(\widehat{Y} = 1) - \mathbb{P}_{X|A=0,Y=0}(\widehat{Y} = 1) \tag{35}$$

$$\text{DEA} = \mathbb{P}(\widehat{Y} \neq Y \mid A = 1) - \mathbb{P}(\widehat{Y} \neq Y \mid A = 0). \tag{36}$$

Specifically, we define $|DEO| \preceq (\alpha_1, \alpha_2)$, which is equivalent to $|\mathbb{P}_{X|A=1,Y=1}(\widehat{Y} = 1) - \mathbb{P}_{X|A=0,Y=1}(\widehat{Y} = 1)| \leq \alpha_1$ and $|\mathbb{P}_{X|A=1,Y=0}(\widehat{Y} = 1) - \mathbb{P}_{X|A=0,Y=0}(\widehat{Y} = 1)| \leq \alpha_2$.

## B.2 FedFaiREE for DDP

**Proposition B.5.** *Under Assumption 3.1, for $a \in \{0,1\}$, consider $k^{*,a} \in \{1,\ldots,n^{*,a}\}$, the corresponding $\hat{k}_i^{*,a}$ for $i \in [S]$ which are $\varepsilon$-approximate ranks and the score-based classifier $\phi(x,a) = \mathbb{1}\{f(x,a) > t_{(k^{*,a})}^{*,a}\}$ . Define*

$$h_{*,a}(\boldsymbol{u}, \boldsymbol{v}) = \mathbb{P}\big(\sum_{i=1}^{S} \pi_i^{*,a} Q(u_i, n_i^{*,a} + 1 - u_i) - \sum_{i=1}^{S} \pi_i^{*,1-a} Q(v_i, n_i^{*,1-a} + 1 - v_i) \geq \alpha\big).$$

*Then we have:*

$$\mathbb{P}(|DDP(\phi)| > \alpha) \leq h_{*,0}(\boldsymbol{M}^{*,0}, \boldsymbol{m}^{*,1}) + h_{*,1}(\boldsymbol{M}^{*,1}, \boldsymbol{m}^{*,0}) \tag{37}$$

*Where $\pi_i^{*,a} = \mathbb{P}(sampling\ x\ from\ client\ i \mid sampling\ x\ with\ sensitive\ attribute A = a)$, $M_i^{*,a} = \min\left(\lceil \hat{k}_i^{*,a} + \varepsilon n_i^{*,a}\rceil, n_i^{*,a} + 1\right)$, $m_i^{*,a} = \max\left(\lceil \hat{k}_i^{*,a} - \varepsilon n_i^{*,a}\rceil, 0\right)$, and $Q(A,B)$ are independent random variables following Beta distribution, $Q(A,B) \sim Beta(A,B)$. Especially, we define $Q(0,B) = 0$ and $Q(A,0) = 1$ for $A, B \neq 0$.*

---

**Algorithm 5** FedFaiREE for DDP

**Input:** Calibration Dataset $D_i = D_i^{0,0} \cup D_i^{0,1} \cup D_i^{1,0} \cup D_i^{1,1}$; pre-trained classifier $\phi_0$ with function f; fairness constraint parameter $\alpha$ ; Confidence level parameter $\beta$; Weights of different clients $\pi$

**Output:** classifier $\hat{\phi}(x,a) = \mathbb{1}\{f(x,a) > t_{(k^{1,a})}^{1,a}\}$

1: **Client Side:**
2: **for** i=1,2,..,S **do**
3:     Score on train data points in $D_i$ and get $T_i^{y,a} = \{t_{i,1}^{y,a}, t_{i,2}^{y,a}, \cdots, t_{i,n_i^{y,a}}^{y,a}\}$
4:     Sort $T_i^{y,a}$
5:     Calculate q-digest of $T_i^{y,a}$ on client $i$
6:     Update digest to server
7: **end for**
8: **Server Side:**
9: Construct $K$ by $K = \{(k^{1,0}, k^{1,1}) | L(\boldsymbol{k}^{1,0}, \boldsymbol{k}^{1,1}) < 1 - \beta\}$, where L is defined by the right-hand side of Inequality 37
10: Select optimal $(k_0, k_1)$ by minimizing equation 9 using estimated values $\hat{p}_a^i = \frac{n_i^{0,a} + n_i^{1,a}}{n_i^{0,0} + n_i^{0,1} + n_i^{1,0} + n_i^{1,1}}$ and $\hat{p}_{Y,a}^i = \frac{n_i^{1,a}}{n_i^{0,a} + n_i^{1,a}}$

---

**Theorem B.6.** *Under Assumption 3.1 and 4.1, given $\alpha' < \alpha$. Suppose $\hat{\phi}$ is the final output of FedFaiREE, we have:*

**(1) Fairness guarantee.** $|DDP(\hat{\phi})| < \alpha$ with probability $(1 - \delta)^N$, where $N$ is the size of the candidate set.

**(2) Accuracy guarantee.** Suppose the density distribution functions of $f^*$ under $A = a, Y = 1$ are continuous. When the input classifier $f$ satisfies $|f(x, a) - f^*(x, a)| \leq \epsilon_0$, for any $\epsilon > 0$ such that $F^*_{(+)}(\epsilon + \gamma\varepsilon) \leq \frac{\alpha - \alpha'}{2} - F^*_{(+)}(2\epsilon_0)$, we have

$$\mathbb{P}(\hat{\phi}(x, a) \neq Y) - \mathbb{P}(\phi^*_{\alpha'}(x, a) \neq Y) \leq 2F^*_{(+)}(2\epsilon_0) + 2F^*_{(+)}(\epsilon + \gamma\varepsilon) + 8\epsilon^2 + 20\epsilon + 2\theta \tag{38}$$

with probability $1 - 4\sum_{a=1}^{1}\sum_{i=1}^{S} e^{-2n_i^{0,a}\epsilon^2} - \prod_{i=1}^{S}\left(1 - F_{i(-)}^{1,0}(2\epsilon)\right)^{n_i^{1,0}} - \prod_{i=1}^{S}\left(1 - F_{i(-)}^{1,1}(2\epsilon)\right)^{n_i^{1,1}} - \delta$, where $\delta = \delta^{1,0}(n^{1,0}) + \delta^{1,1}(n^{1,1})$, $\theta$ is defined in Proposition 3.5 and the definition of $F_{(+)}$ and $F_{(-)}$ are shown in Lemma A.4

## B.3 FedFaiREE for DPE

**Proposition B.7.** Under Assumption 3.1, for $a \in \{0, 1\}$, consider $k^{0,a} \in \{1, \ldots, n^{0,a}\}$, the corresponding $\hat{k}_i^{0,a}$ for $i \in [S]$ which are $\varepsilon$-approximate ranks and the score-based classifier $\phi(x, a) = \mathbb{1}\{f(x, a) > t_{(k^{0,a})}^{0,a}\}$. Define

$$h_{y,a}(\boldsymbol{u}, \boldsymbol{v}) = \mathbb{P}(\sum_{i=1}^{S}\pi_i^{y,a}Q(u_i, n_i^{y,a} + 1 - u_i) - \sum_{i=1}^{S}\pi_i^{y,1-a}Q(v_i, n_i^{y,1-a} + 1 - v_i) \geq \alpha).$$

Then we have:
$$\mathbb{P}(|DPE(\phi)| > \alpha) \leq h_{0,1}(\boldsymbol{M}^{0,1}, \boldsymbol{m}^{0,0}) + h_{0,0}(\boldsymbol{M}^{0,0}, \boldsymbol{m}^{0,0}) \tag{39}$$
where $M_i^{0,a} = \lceil\hat{k}_i^{0,a} + \varepsilon n_i^{0,a}\rceil$, $m_i^{0,a} = \lceil\hat{k}_i^{0,a} - \varepsilon n_i^{0,a}\rceil$,

$$\pi_i^{y,a} = \mathbb{P}(sampling\ x\ from\ client\ i\ |\ sampling\ x\ with\ label\ Y = y\ and\ A = a),$$

and $Q(A, B)$ are independent random variables following Beta distribution, $Q(A, B) \sim Beta(A, B)$.

**Theorem B.8.** Under Assumption 3.1 and 4.1, given $\alpha' < \alpha$. Suppose $\hat{\phi}$ is the final output of FedFaiREE, we have:

**(1) Fairness guarantee.** $|DPE(\hat{\phi})| < \alpha$ with probability $(1 - \delta)^N$, where $N$ is the size of the candidate set.

**(2) Accuracy guarantee.** Suppose the density distribution functions of $f^*$ under $A = a, Y = 1$ are continuous. When the input classifier $f$ satisfies $|f(x, a) - f^*(x, a)| \leq \epsilon_0$, for any $\epsilon > 0$ such that $F^*_{(+)}(\epsilon + \gamma\varepsilon) \leq \frac{\alpha - \alpha'}{2} - F^*_{(+)}(2\epsilon_0)$, we have

$$\mathbb{P}(\hat{\phi}(x, a) \neq Y) - \mathbb{P}(\phi^*_{\alpha'}(x, a) \neq Y) \leq 2F^*_{(+)}(2\epsilon_0) + 2F^*_{(+)}(\epsilon + \gamma\varepsilon) + 8\epsilon^2 + 20\epsilon + 2\theta \tag{40}$$

with probability $1 - 4\sum_{a=1}^{1}\sum_{i=1}^{S} e^{-2n_i^{0,a}\epsilon^2} - \prod_{i=1}^{S}\left(1 - F_{i(-)}^{1,0}(2\epsilon)\right)^{n_i^{1,0}} - \prod_{i=1}^{S}\left(1 - F_{i(-)}^{1,1}(2\epsilon)\right)^{n_i^{1,1}} - \delta$, where $\delta = \delta^{1,0}(n^{1,0}) + \delta^{1,1}(n^{1,1})$, $\theta$ is defined in Proposition 3.5 and the definition of $F_{(+)}$ and $F_{(-)}$ are shown in Lemma A.4

---

**Algorithm 6** FedFaiREE for DPE

---

**Input:** Calibration Dataset $D_i = D_i^{0,0} \cup D_i^{0,1} \cup D_i^{1,0} \cup D_i^{1,1}$; pre-trained classifier $\phi_0$ with function f; fairness constraint parameter $\alpha$ ; Confidence level parameter $\beta$; Weights of different clients $\pi$
**Output:** classifier $\hat{\phi}(x,a) = \mathbb{1}\{f(x,a) > t_{(k^{1,a})}^{1,a}\}$

1: **Client Side:**
2: **for** i=1,2,..,$S$ **do**
3:     Score on train data points in $D_i$ and get $T_i^{y,a} = \{t_{i,1}^{y,a}, t_{i,2}^{y,a}, \cdots, t_{i,n_i^{y,a}}^{y,a}\}$
4:     Sort $T_i^{y,a}$
5:     Calculate q-digest of $T_i^{y,a}$ on client $i$
6:     Update digest to server
7: **end for**
8: **Server Side:**
9: Construct $K$ by $K = \{(k^{1,0}, k^{1,1}) | L(\boldsymbol{k}^{1,0}, \boldsymbol{k}^{1,1}) < 1-\beta\}$, where L is defined by the right-hand side of Inequality 39
10: Select optimal $(k_0, k_1)$ by minimizing equation 9 using estimated values $\hat{p}_a^i = \frac{n_i^{0,a}+n_i^{1,a}}{n_i^{0,0}+n_i^{0,1}+n_i^{1,0}+n_i^{1,1}}$ and $\hat{p}_{Y,a}^i = \frac{n_i^{1,a}}{n_i^{0,a}+n_i^{1,a}}$

---

## B.4   FedFaiREE for DEA

**Proposition B.9.** *Under Assumption 3.1, for $a \in \{0,1\}$, consider $k^{y,a} \in \{1,\ldots,n^{y,a}\}$, the corresponding $\hat{k}_i^{y,a}$ for $i \in [S]$ which are $\varepsilon$-approximate ranks and the score-based classifier $\phi(x,a) = \mathbb{1}\{f(x,a) > t_{(k^{1,a})}^{1,a}\}$ . Define*

$$h_{*,a}(\boldsymbol{u}^1, \boldsymbol{u}^0, \boldsymbol{v}^1, \boldsymbol{v}^0) = \mathbb{P}\Bigg( p_{y,a} - p_{y,1-a} - p_{y,a} \sum_{i=1}^{S} \pi_i^{1,a} Q\left(u_i^1, n_i^{1,a} + 1 - u_i^1\right)$$

$$+ (1 - p_{y,a}) \sum_{i=1}^{S} \pi_i^{0,a} Q\left(u_i^0, n_i^{0,a} + 1 - u_i^0\right)$$

$$+ p_{y,1-a} \sum_{i=1}^{S} \pi_i^{1,1-a} Q\left(v_i^1, n_i^{1,1-a} + 1 - v_i^1\right)$$

$$- (1 - p_{y,1-a}) \sum_{i=1}^{S} \pi_i^{0,1-a} Q\left(v_i^0, n_i^{0,1-a} + 1 - v_i^0\right) \geq \alpha \Bigg).$$

*Then we have:*

$$\mathbb{P}(|DPE(\phi)| > \alpha) \leq h_{*,1}(\boldsymbol{m}^{1,1}, \boldsymbol{M}^{0,1}, \boldsymbol{M}^{1,0}, \boldsymbol{m}^{0,0}) + h_{*,0}(\boldsymbol{m}^{1,0}, \boldsymbol{M}^{0,0}, \boldsymbol{M}^{1,1}, \boldsymbol{m}^{0,1}) \tag{41}$$

*where $M_i^{0,a} = \lceil \hat{k}_i^{0,a} + \varepsilon n_i^{0,a} \rceil$, $m_i^{0,a} = \lceil \hat{k}_i^{0,a} - \varepsilon n_i^{0,a} \rceil$,*

$$\pi_i^{y,a} = \mathbb{P}(sampling\ x\ from\ client\ i \mid\ sampling\ x\ with\ label\ Y = y\ and\ A = a),$$

*and $Q(A,B)$ are independent random variables following Beta distribution, $Q(A,B) \sim Beta(A,B)$.*

**Theorem B.10.** *Under Assumption 3.1 and 4.1, given $\alpha' < \alpha$. Suppose $\hat{\phi}$ is the final output of FedFaiREE, we have:*

*(1) Fairness guarantee. $|DEA(\hat{\phi})| < \alpha$ with probability $(1-\delta)^N$, where $N$ is the size of*

**Algorithm 7** FedFaiREE for DEA

---

**Input:** Calibration Dataset $D_i = D_i^{0,0} \cup D_i^{0,1} \cup D_i^{1,0} \cup D_i^{1,1}$; pre-trained classifier $\phi_0$ with function f; fairness constraint parameter $\alpha$; Confidence level parameter $\beta$; Weights of different clients $\pi$

**Output:** classifier $\hat{\phi}(x,a) = \mathbb{1}\{f(x,a) > t_{(k^{1,a})}^{1,a}\}$

1: **Client Side:**
2: **for** i=1,2,..,$S$ **do**
3:     Score on train data points in $D_i$ and get $T_i^{y,a} = \{t_{i,1}^{y,a}, t_{i,2}^{y,a}, \cdots, t_{i,n_i^{y,a}}^{y,a}\}$
4:     Sort $T_i^{y,a}$
5:     Calculate q-digest of $T_i^{y,a}$ on client $i$
6:     Update digest to server
7: **end for**
8: **Server Side:**
9: Construct $K$ by $K = \{(k^{1,0}, k^{1,1}) | L(\boldsymbol{k}^{1,0}, \boldsymbol{k}^{1,1}) < 1 - \beta\}$, where L is defined by the right-hand side of Inequality 41
10: Select optimal $(k_0, k_1)$ by minimizing equation 9 using estimated values $\hat{p}_a^i = \frac{n_i^{0,a} + n_i^{1,a}}{n_i^{0,0} + n_i^{0,1} + n_i^{1,0} + n_i^{1,1}}$ and $\hat{p}_{Y,a}^i = \frac{n_i^{1,a}}{n_i^{0,a} + n_i^{1,a}}$

---

*the candidate set.*

*(2) **Accuracy guarantee.** Suppose the density distribution functions of $f^*$ under $A = a, Y = 1$ are continuous. When the input classifier $f$ satisfies $|f(x,a) - f^*(x,a)| \leq \epsilon_0$, for any $\epsilon > 0$ such that $F_{(+)}^*(\epsilon + \gamma\varepsilon) \leq \frac{\alpha - \alpha'}{2} - F_{(+)}^*(2\epsilon_0)$, we have*

$$\mathbb{P}(\hat{\phi}(x,a) \neq Y) - \mathbb{P}(\phi_{\alpha'}^*(x,a) \neq Y) \leq 2F_{(+)}^*(2\epsilon_0) + 2F_{(+)}^*(\epsilon + \gamma\varepsilon) + 8\epsilon^2 + 20\epsilon + 2\theta \tag{42}$$

*with probability $1 - 4\sum_{a=1}^1 \sum_{i=1}^S e^{-2n_i^{0,a}\epsilon^2} - \prod_{i=1}^S \left(1 - F_{i(-)}^{1,0}(2\epsilon)\right)^{n_i^{1,0}} - \prod_{i=1}^S \left(1 - F_{i(-)}^{1,1}(2\epsilon)\right)^{n_i^{1,1}} - \delta$, where $\delta = \delta^{1,0}(n^{1,0}) + \delta^{1,1}(n^{1,1})$, $\theta$ is defined in Proposition 3.5 and the definition of $F_{(+)}$ and $F_{(-)}$ are shown in Lemma A.4*

## B.5    Connection with Fairness Metrics in (Hu et al., 2022) and (Papadaki et al., 2022)

Hu et al. (2022) introduces several group fairness metrics as follow:

**Definition B.11.** A classifier h satisfies Bounded Group Loss (BGL) at level $\zeta$ under distribution $\mathcal{D}$ if for all $a \in A$, we have $\mathbb{E}[l(h(x), y) \mid A = a] \leq \zeta$.

**Definition B.12.** A classifier $h$ satisfies Conditional Bounded Group Loss (CBGL) for $y \in Y$ at level $\zeta_y$ under distribution $\mathcal{D}$ if for all $a \in A$, we have $\mathbb{E}[l(h(x), y) \mid A = a, Y = y] \leq \zeta_y$.

When considering y as a binary variable and the loss function l being the 0-1 loss function, BGL is equivalent to

$$\mathbb{P}[\hat{y} \neq y| \mid A = a] \leq \zeta,$$

holding for any a, whereas Demographic Parity refers to

$$\mathbb{P}[\hat{y} \neq y| \mid A = 0] = \mathbb{P}[\hat{y} \neq y| \mid A = 1].$$

In this context, BGL can be understood as a relaxation of Demographic Parity.

Similarly, when considering y as a binary variable and the loss function l being the 0-1 loss function, CBGL is equivalent to

$$\mathbb{P}[\hat{y} \neq y| \mid A = a, Y = y] \leq \zeta_y,$$

holding for any a, whereas Equalized Odds refers to

$$\mathbb{P}[\hat{y} \neq y| \mid A = 0, Y = y] = \mathbb{P}[\hat{y} \neq y| \mid A = 1, Y = y].$$

In this context, CBGL can be understood as a relaxation of Equalized Odds.

According to (Hu et al., 2022), the metric that Papadaki et al. (2022) considers is equivalent to following definition.

**Definition B.13.** FedMinMax (Papadaki et al., 2022) aims to solve for the following objective: $\min_h \max_{\boldsymbol{\lambda} \in \mathbb{R}_+^{|A|}, \|\boldsymbol{\lambda}\|_1 = 1} \sum_{a \in A} \boldsymbol{\lambda}_a \mathbf{r}_a(h)$, where

$$\mathbf{r}_a(h) := \sum_{k=1}^{K} \mathbf{r}_{a,k}(h) = \sum_{k=1}^{K} \left( 1/m_a \sum_{a_{k,i}=a} l\left(h\left(x_{k,i}\right), y_{k,i}\right) \right),$$

$K$ stands for client number and $m_a$ stands for numbers of points with attribute $a$.

Similarly, this can be understood as a relaxation of Demographic Parity in the context of considering y as a binary variable and the loss function l being the 0-1 loss function.

## C  Experiment Details

This section provides additional implementation details and supplementary experimental analyses supporting the numerical results in Section 5. Appendix C.1 describes the simplified candidate-set search used in FedFaiREE to reduce the computational cost of selecting feasible rank pairs. Appendix C.2 presents the experimental protocol, including the federated backbone models, hyperparameter selection, dataset partitioning, Q-digest settings, and computational resources. Appendix C.3 reports additional repeated-run results, including standard deviations, and studies the sensitivity of FedFaiREE to the fairness tolerance $\alpha$ and confidence level $\beta$. Appendix C.4 compares FedFaiREE with local FaiREE, centralized FaiREE, and centralized FedFaiREE. Appendix C.5 studies the effect of the quality of the input scoring function $f$, showing how scorer quality affects predictive accuracy while the fairness-control behavior remains stable. Appendix C.6 investigates the robustness of FedFaiREE under partial client participation. Finally, Appendix C.7 reports additional results on the German dataset to further evaluate the generality of the proposed method.

## C.1 Simplified selection in candidate set construction

To further simplify the candidate set selection, similar to FaiREE (Li et al., 2022), we note that, by Lemma A.2, if we assume our input classifier $f$ is similar to $f^*$, we have

$$t_a = \frac{p_a p_{Y,a}}{2 p_a p_{Y,a} + (1 - 2a) t^{\star}_{E,\alpha}}, \tag{43}$$

which means

$$t^{\star}_{E,\alpha} = \frac{p_a p_{Y,a} - 2 p_a p_{Y,a} t_a}{(1 - 2a) t_a} \tag{44}$$

Therefore, bringing equation 44 ($a = 0$) into equation 43 ($a = 1$), we have

$$t_0 = \frac{p_0 p_{Y,0}}{2 p_0 p_{Y,0} + 2 p_1 p_{Y,1} - p_1 p_{Y,1} / t_1} \tag{45}$$

This inspired us that we could further simplify the construction of candidate set K by replacing equation 8 with

$$K = \{(k^{1,0}, k^{1,1}) | L(\boldsymbol{k}^{1,0}, \boldsymbol{k}^{1,1}) < 1 - \beta, k^{1,0} = \mu(k^{1,1})\}, \tag{46}$$

Where $\mu(k_1) = \arg\min_{k_0} \frac{p_0 p_{Y,0}}{2 p_0 p_{Y,0} + 2 p_1 p_{Y,1} - p_1 p_{Y,1} / \hat{t}_{k_1}}$

## C.2 Model Details and Hyperparameter Selection

We employed several existing Federated Learning models in the experiment, and their detailed information is listed as follows:

1. FedAvg (McMahan et al., 2017): FedAvg is a fundamental Federated Learning model that serves as the foundational baseline for our experiments. It operates by computing model updates on each client's local data and then aggregates these updates on a central server through averaging. FedAvg doesn't specifically address fairness concerns but is crucial for benchmarking purposes.

2. FedFB (Zeng et al., 2021): FedFB is a novel framework designed for fairness-aware Federated Learning. Drawing inspiration from FairBatch, a fairness algorithm for centralized data, FedFB extends this concept to the Federated Learning setting. It incorporates both local debiasing and global reweighting for each client within the framework to achieve fairness objectives.

3. FairFed (Ezzeldin et al., 2023): FairFed is another innovative framework for fairness-aware Federated Learning. It employs a unique approach to improving fairness by reweighting clients based on updated local fairness indicators during each epoch. This allows FairFed to combine multiple local debiasing methods effectively.

To compare performance in terms of DEOO, we selected FedFB with respect to Equal Opportunity (EO) as presented in Zeng et al. (2021), and FairFed-FB-EO from FairFed as introduced in Ezzeldin et al. (2023). These are specific models within the FedFB and FairFed frameworks that are designed for DEOO.

We also note that there are concerns raised by the fairness community regarding the COMPAS dataset underscore crucial complexities within algorithmic fairness research (Bao et al., 2021). While Risk Assessment Instrument (RAI) datasets like COMPAS serve as prevalent benchmarks, their oversimplification of the intricate dynamics within real-world criminal justice processes poses significant challenges. Measurement biases and errors inherent in pretrial RAI datasets limit the direct translation of fairness claims to actual outcomes within the criminal justice system. Additionally, the technical focus on these data as a benchmark sometimes ignores the contextual grounding necessary for working with RAI datasets. Ethical reflection within socio-technical systems further highlights the necessity of acknowledging and grappling with the limitations and complexities inherent in RAI datasets.

Additionally, the hyperparameter selection ranges for each model are shown in Table 6.

We further present a data split sample in Table 7, where random seed was set to be 0.

In our experiments, the main computational resources utilized were mobile versions of the 4080 and 4090 GPUs, as well as a T4 GPU on the server. After processing, the input size of the CelebA dataset was approximately 2.31 GB for 64*64 images, and for acsincome, it was around 135 MB. The storage occupied by other datasets did not exceed 10 MB.Due to limited computing resources, we conducted only eight experiments on CelebA, and therefore, we do not provide the 95% quantile of DEOO. Specifically, the absence of FedFB results in the Table 2 is due to insufficient GPU memory.

## C.3   Detailed experimental Set-up for the real data analysis

In this subsection, we present a more detailed analysis of the experimental results from Section 5. Table 8 and Table 9 respectively illustrate the variances in the results obtained from the Adult dataset and the Compas dataset.

Table 10 shows the result on adult with parameter for Dirichlet distribution=10. Moreover, we present an analysis of the impact of parameter variations on the experimental results. We consider two parameters——the fairness constraint, $\alpha$, and the confidence coefficient, $\beta$, separately. Figure 3 and 4 shows the result on Adult dataset and Compas dataset, respectively.

Figures 3 and 4 illustrate the sensitivity of FedFaiREE to the fairness tolerance parameter $\alpha$ and the confidence-level parameter $\beta$ on the Adult and Compas datasets, respectively. Overall, the results show a clear and reasonable fairness-accuracy trade-off. As $\alpha$ increases, the fairness constraint becomes less restrictive, enlarging the feasible candidate set. Consequently, FedFaiREE can select classifiers with higher predictive accuracy. At the same time, across different choices of $\alpha$, the empirical $|DEOO|_{95}$ remains consistently controlled below the prescribed tolerance level, demonstrating the robustness of FedFaiREE in satisfying the target high-probability fairness constraint. In contrast, when $\beta$ increases, FedFaiREE enforces the constraint with a higher confidence requirement, which makes the candidate selection more conservative. This conservativeness slightly decreases accuracy but also leads to stronger fairness control, as reflected by smaller $|DEOO|_{95}$ and $DEOO$ mean.

**Table 6:** Hyperparameter Selection Ranges

| Model | Hyperparameter | Ranges |
|---|---|---|
| General | Learning rate | $\{0.001, 0.005, 0.01\}$ |
| | Global round | $\{5, 10, 20, 30, 40, 50, 80\}$ |
| | Local round | $\{5, 10\}$ |
| | Local batch size | $\{16, 32, 64, 128\}$ |
| | Hidden layer | $\{5, 10, 50\}$ for Adult, Compas, ACSIncome, German |
| | Optimizer | $\{$Adam, Sgd$\}$ |
| | Fraction | $\{1\}$ |
| | Parameter for Dirichlet distribution | $\{1\}$ for Adult, $\{10\}$ for Compas and German |
| | Number of Clients | $\{100\}$ for Adult, $\{10\}$ for Compas and German, $\{50\}$ for ACSIncome, $\{508\}$ for CelebA |
| | Sensitive Group | Female for Adult, Compas and German, Non-white for ACSIncome, Male for CelebA |
| FedFaiREE | Confidence level | $\{95\%\}$ |
| Qdigest | Accuracy | $\{1/2^7\}$ for Adult, ACSIncome, German and CelebA, $\{1/2^{10}\}$ for Compas |
| | Compression factor | $\{300\}$ for Adult, ACSIncome, German and CelebA, $\{150\}$ for Compas |
| FedFB | Step size ($\alpha$) | $\{0.005, 0.01, 0.05\}$ |
| FairFed | Global step size ($\beta$) | $\{0.005, 0.01, 0.05\}$ |
| | Local debiasing step size ($\alpha$) | $\{0.005, 0.01, 0.05\}$ |

Table 7: **Heterogeneous data distribution on the sensitive attribute.** The client index is sorted by number of Male.

| Minimum ten clients | | | Maximum ten clients | | |
|---|---|---|---|---|---|
| Client id | Male | Female | Client id | Male | Female |
| 1 | 6 | 41 | 91 | 738 | 118 |
| 2 | 6 | 117 | 92 | 863 | 49 |
| 3 | 6 | 297 | 93 | 880 | 52 |
| 4 | 13 | 35 | 94 | 956 | 147 |
| 5 | 20 | 310 | 95 | 961 | 50 |
| 6 | 22 | 120 | 96 | 1101 | 35 |
| 7 | 24 | 234 | 97 | 1245 | 102 |
| 8 | 30 | 70 | 98 | 1250 | 31 |
| 9 | 32 | 124 | 99 | 1277 | 180 |
| 10 | 33 | 26 | 100 | 1480 | 24 |

Table 8: **Results with standard deviation on Adult.**

| Model | **FedFaiREE** | $\alpha$ | $\overline{ACC}$ | $\overline{|DEOO|}$ | $|DEOO|_{95}$ |
|---|---|---|---|---|---|
| | | | **Adult** | | |
| **FedAvg** | ✗ | / | 0.844 (0.003) | 0.131 (0.030) | 0.178 |
| | ✓ | 0.10 | 0.843 (0.003) | **0.038** (0.026) | **0.083** |
| **FedFB** | ✗ | / | 0.850 (0.003) | 0.057 (0.034) | 0.117 |
| | ✓ | 0.10 | 0.850 (0.003) | **0.036** (0.025) | **0.083** |
| **FairFed** | ✗ | / | 0.842 (0.003) | 0.069 (0.034) | 0.118 |
| | ✓ | 0.10 | 0.841 (0.003) | **0.037** (0.026) | **0.081** |

## C.4 Compared to local FaiREE and centralized FedFaiREE

We have added two experiments on Adult compared to local FaiREE and centralized FedFaiREE. The results are shown in Table 11. Here, "local FaiREE" refers to each client using FaiREE individually (without collaboration), "FaiREE" refers to FaiREE applied when all data is centralized, and "Centralized FedFaiREE" refers to FedFaiREE applied when all data is centralized.

Overall, Local FaiREE may be limited by the size of client data. Although threshold estimation based on client data can ensure fairness, there is a significant loss in accuracy. FaiREE

Table 9: **Results with standard deviation on Compas.**

| Model | **FedFaiREE** | $\alpha$ | $\overline{ACC}$ | $\overline{|DEOO|}$ | $|DEOO|_{95}$ |
|---|---|---|---|---|---|
| | | | **Compas** | | |
| **FedAvg** | ✗ | / | 0.662 (0.011) | 0.126 (0.056) | 0.223 |
| | ✓ | 0.15 | 0.659 (0.010) | **0.051** (0.044) | **0.137** |
| **FedFB** | ✗ | / | 0.642 (0.011) | 0.107 (0.043) | 0.174 |
| | ✓ | 0.15 | 0.641 (0.010) | **0.062** (0.040) | **0.125** |
| **FairFed** | ✗ | / | 0.648 (0.012) | 0.097 (0.047) | 0.166 |
| | ✓ | 0.15 | 0.645 (0.011) | **0.047** (0.036) | **0.114** |

Table 10: **Results on Adult with Parameter for Dirichlet distribution=10.**

| Model | **FedFaiREE** | $\alpha$ | $\overline{ACC}$ | $\overline{|DEOO|}$ | $|DEOO|_{95}$ |
|---|---|---|---|---|---|
| **FedAvg** | ✗ | / | 0.844 (0.004) | 0.127 (0.032) | 0.184 |
| | ✓ | 0.10 | 0.843 (0.003) | **0.029** (0.027) | **0.091** |
| **FedFB** | ✗ | / | 0.845 (0.003) | 0.057 (0.034) | 0.117 |
| | ✓ | 0.10 | 0.845 (0.003) | **0.036** (0.025) | **0.083** |
| **FairFed** | ✗ | / | 0.839 (0.004) | 0.081 (0.033) | 0.138 |
| | ✓ | 0.10 | 0.838 (0.004) | **0.027** (0.025) | **0.073** |

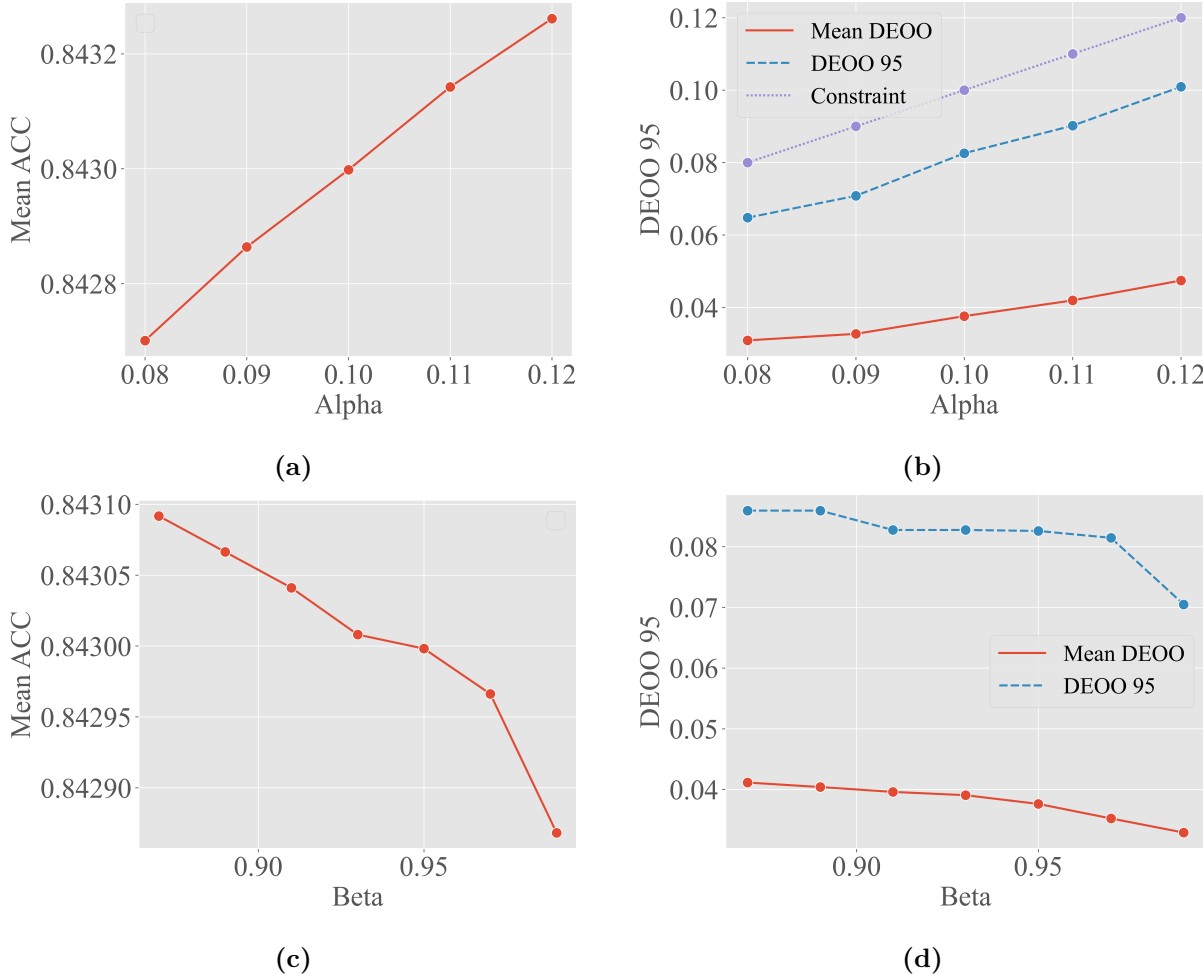

(a)

(b)

(c)

(d)

Figure 3: **The changes of accuracy, $\overline{|DEOO|}$ and $|DEOO|_{95}$ with respect to $\alpha$ and $\beta$ on Adult.** The other parameters of the experiment are consistent with those in Table 1.

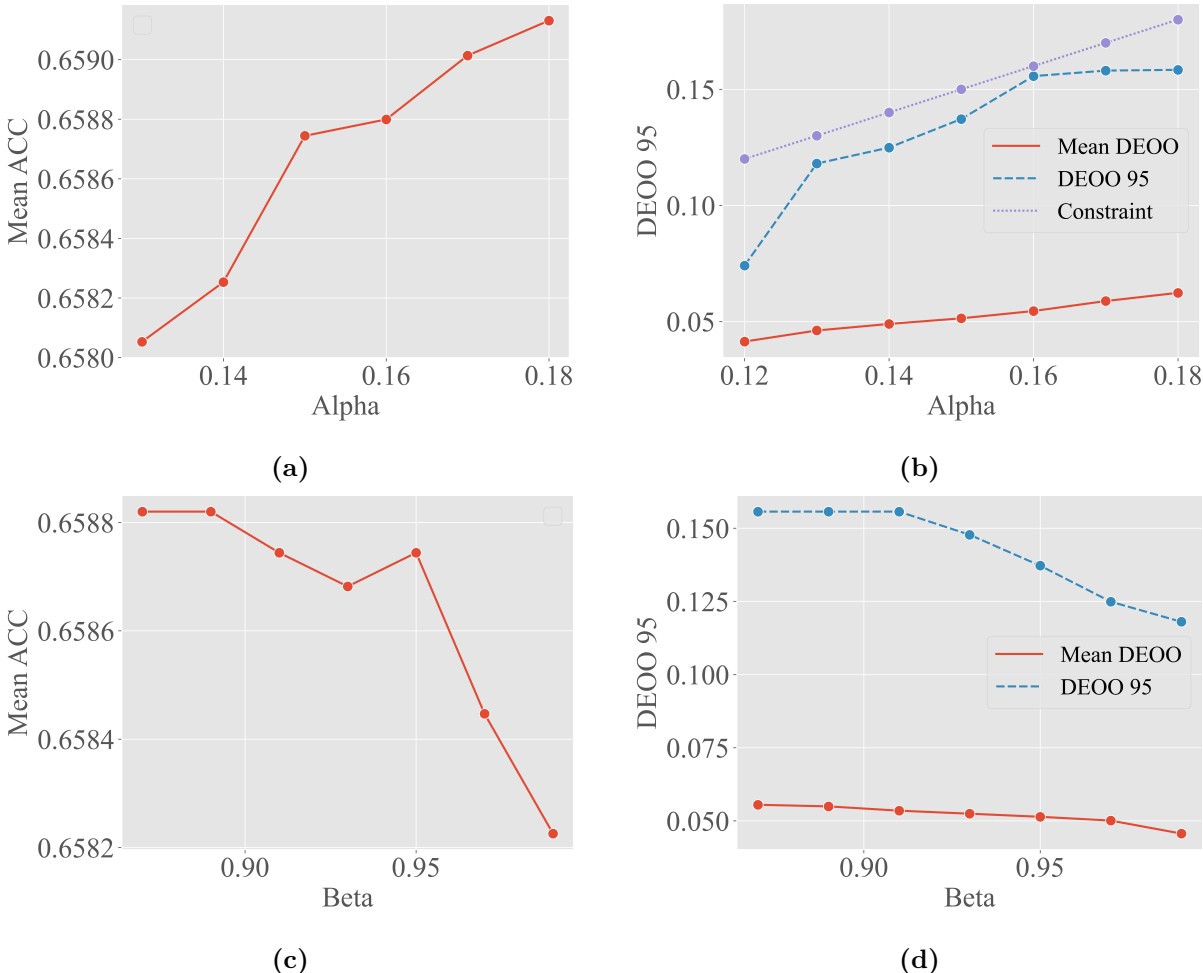

Figure 4: **The changes of accuracy, $\overline{|DEOO|}$ and $|DEOO|_{95}$ with respect to $\alpha$ and $\beta$ on Compas.** The other parameters of the experiment are consistent with those in Table 1.

may be influenced by heterogeneity and thus may not guarantee fairness effectively. In contrast, FedFaiREE, by considering local statistics, can better ensure fairness in environments with heterogeneity. Although Centralized FedFaiREE shows a slight improvement in accuracy compared to FedFaiREE while maintaining fairness, the overall improvement is not significant (up to the fourth decimal place). Therefore, the performance of FedFaiREE, which introduces approximate quantiles to reduce communication costs, is already quite satisfactory.

## C.5 Experiments on Different Goodness Scoring Function.

We note that the fairness bound does not depend on the goodness of $f$, while the accuracy bound does depend on the goodness of $f$ through the parameter $\epsilon_0$ in Theorem 4.2. To further elucidate this, we have conducted experiments comparing different goodness level scoring functions. The results are shown in Table 12.

The results in Table 12 further provide empirical evidence for the different roles of the scoring function quality in accuracy and fairness control. As the number of FedAvg training rounds increases from 5 to 20, the predictive accuracy of the original scorer improves, and the accuracy after applying FedFaiREE also increases from 0.784 to 0.835. This indicates that a better scoring

Table 11: Comparison of Local FaiREE, FaiREE, FedFaiREE and Centralizd FedFaiREE.

| Model | Method | $\alpha$ | $\overline{ACC}$ | $\overline{|DEOO|}$ | $|DEOO|_{95}$ |
|---|---|---|---|---|---|
| FedAvg | / | / | 0.844 | 0.131 | 0.178 |
| | Local FaiREE | 0.10 | 0.776 | 0.038 | 0.092 |
| | FaiREE | 0.10 | 0.844 | 0.071 | 0.128 |
| | FedFaiREE | 0.10 | 0.843 | 0.038 | 0.083 |
| | Centralizd FedFaiREE | 0.10 | 0.844 | 0.042 | 0.087 |
| FedFB | / | / | 0.850 | 0.057 | 0.117 |
| | Local FaiREE | 0.10 | 0.771 | 0.031 | 0.075 |
| | FaiREE | 0.10 | 0.850 | 0.055 | 0.109 |
| | FedFaiREE | 0.10 | 0.850 | 0.036 | 0.083 |
| | Centralizd FedFaiREE | 0.10 | 0.850 | 0.040 | 0.093 |
| FairFed | / | / | 0.842 | 0.069 | 0.118 |
| | Local FaiREE | 0.10 | 0.775 | 0.031 | 0.069 |
| | FaiREE | 0.10 | 0.842 | 0.066 | 0.112 |
| | FedFaiREE | 0.10 | 0.841 | 0.037 | 0.081 |
| | Centralizd FedFaiREE | 0.10 | 0.842 | 0.043 | 0.086 |

Table 12: Results with scoring function f using FedAvg under different global training rounds on Adult. The notation meanings align with those in Table 1.

| Model | Method | $\alpha$ | $\overline{ACC}$ | $\overline{|DEOO|}$ | $|DEOO|_{95}$ |
|---|---|---|---|---|---|
| 5 Rounds | / | / | 0.786 | 0.185 | 0.335 |
| | FedFaiREE | 0.10 | 0.784 | 0.033 | 0.072 |
| 10 Rounds | / | / | 0.826 | 0.278 | 0.413 |
| | FedFaiREE | 0.10 | 0.826 | 0.036 | 0.084 |
| 20 Rounds | / | / | 0.836 | 0.189 | 0.241 |
| | FedFaiREE | 0.10 | 0.835 | 0.035 | 0.087 |

function mainly improves the utility of the final classifier. In contrast, the fairness performance of FedFaiREE remains relatively stable across different scorer qualities. The average $\overline{|DEOO|}$ stays within a narrow range of 0.033–0.036, and $|DEOO|_{95}$ remains below the target tolerance $\alpha = 0.10$ in all cases. This observation is consistent with Theorem 4.2.

## C.6 Experiments on Different Levels of Client Participation.

In real-world scenarios, we often encounter situations where it's not possible to access the entire dataset. In this section, we provide a possible solution — one can consider introducing a hypothesis space denoted as $H(\pi)$ to model the range of $\pi$ and incorporate $\max_{\pi \in H(\pi)}$ into equations 8 and 9. To further illustrate the effectiveness of this method, we conducted the following experiment. The dataset's partitioning and model parameters were kept the same as those in the experiment on the adult dataset. However, we introduced an independent participation rate for each client, following a uniform distribution on [0.3, 0.7]. Subsets were randomly sampled from the original training set based on the participation rate to form a new training set. The results of the experiment are shown in Table 13. Overall, our method still effectively controlled the averages and 95th quantile of DEOO ($\overline{|DEOO|}$ and $|DEOO|_{95}$), while maintaining a high

level of accuracy.

Table 13: Results with random participation rate on Adult. In this Experiment, participation rates of clients are following uniform distribution on [0.3, 0.7]. The notation meanings align with those in Table 1.

| Model | Method | $\alpha$ | $\overline{ACC}$ | $\overline{|DEOO|}$ | $|DEOO|_{95}$ |
|-------|--------|----------|------------------|---------------------|---------------|
| FedAvg | / | / | 0.843 | 0.130 | 0.180 |
|  | FedFaiREE | 0.10 | 0.842 | 0.032 | 0.080 |
| FedFB | / | / | 0.848 | 0.045 | 0.102 |
|  | FedFaiREE | 0.10 | 0.847 | 0.033 | 0.071 |
| FairFed | / | / | 0.847 | 0.075 | 0.152 |
|  | FedFaiREE | 0.10 | 0.846 | 0.045 | 0.102 |

## C.7   Experiment on German

We also run experiments on German dataset(Dua et al., 2017), which involves predicting whether a bank account holder's credit is good or bad. The protected attribute in this dataset is gender. Results are shown in Table 14.

Table 14: Results on German.The notation meanings align with those in Table 1.

| Model | FedFaiREE | $\alpha$ | $\overline{ACC}$ | $\overline{|DEOO|}$ | $|DEOO|_{95}$ |
|-------|-----------|----------|------------------|---------------------|---------------|
| FedAvg | ✗ | / | 0.748 | 0.067 | 0.144 |
|  | ✓ | 0.10 | 0.741 | 0.048 | 0.101 |
| FedFB | ✗ | / | 0.721 | 0.059 | 0.125 |
|  | ✓ | 0.10 | 0.718 | 0.028 | 0.079 |
| FairFed | ✗ | / | 0.731 | 0.060 | 0.130 |
|  | ✓ | 0.10 | 0.725 | 0.042 | 0.101 |

## D   Comparison to FaiREE (Li et al., 2022) and other related works

Regarding the differences between FedFaiREE and FaiREE, several pivotal distinctions become evident. Primarily, FedFaiREE demonstrates superior adaptability for practical applications. Notably, it incorporates mechanisms to handle label shift scenarios, ensuring model robustness within such distributions, as elucidated in Section 5.1. Furthermore, it's worth noting that FedFaiREE extends considerations to encompass multiple sensitive groups and multiple labels, aligning more closely with practical real-world application scenarios, as discussed in Appendix D.

Another critical difference lies in the setting: FaiREE operates in a centralized environment, assuming homogeneous data across all clients. In contrast, FedFaiREE is expressly tailored for decentralized settings, acknowledging client heterogeneity and effectively addressing the challenges stemming from diverse data distributions and sizes across clients. This tailored approach

significantly enhances its adaptability and robustness across various scenarios.

Lastly, while FaiREE relies on specific centralized quantile estimation methods, FedFaiREE adopts approximate quantiles. This adaptation not only facilitates adaptation to distributed data but also fortifies the method's robustness and adaptability.

## D.1 Comparison to other related works

Differences between FedFaiREE and other fair federated learning methods lie in their approach to addressing fairness concerns. Many methods, akin to this paper, extend the principles of centralized machine learning to decentralized settings, such as FedFB(Zeng et al., 2021), Fed-MinMax(Papadaki et al., 2022), PFFL(Hu et al., 2022), and others. These methods primarily focus on introducing fairness penalties in the objective functions and incorporate client reweighting schemes and terms (in objective functions) reweighting schemes that consider global or local fairness. The key divergence between our approach and these methods is that the latter typically converge and provide fairness guarantees only in large-sample scenarios, lacking assurances for fairness in small-sample situations, especially under distribution-free assumptions. Empirical results from Table 1 in this paper demonstrate that compared to FedFaiREE, methods like FedFB, FairFed are not as effective in controlling fairness in small-sample scenarios. Furthermore, as these methods are predominantly in-processing techniques, while FedFaiREE falls under post-processing methods, there is a potential for further integration to achieve improved fairness guarantees as shown in our experiments. Moreover, another significant characteristic of FedFaiREE is its capability to adjust the trade-off between fairness and accuracy according to specific fairness constraints. This control capacity has been demonstrated in numerous experiments, showcasing an ability that other methods lack.

# E    Discussion on Continuity

In the main theoretical development, the order-statistic argument is presented under a continuity condition on the score distribution. This condition is used to invoke the probability integral transform and to obtain the Beta distribution of transformed order statistics. In practice, however, score functions may produce ties, especially when the model output is discrete, rounded, or compressed by a quantile sketch. We now describe a standard randomized tie-breaking convention under which the same finite-sample fairness guarantee remains valid without requiring the original score distribution to be continuous.

For each score $Score = f(X, A)$, we introduce an auxiliary random variable $U \sim \text{Unif}(0, 1)$, independent of all data and all other sources of randomness. We then order observations lexicographically by the augmented score $(Score, U)$. That is, $(Score_1, U_1) \preceq (Score_2, U_2)$ if either $Score_1 < Score_2$, or $Score_1 = Score_2$ and $U_1 \leq U_2$. This rule breaks ties randomly while preserving the original ordering whenever the scores are distinct.

Equivalently, if the group-specific threshold selected by FedFaiREE is $\lambda_a$, the deployed classifier can be written as

$$\widehat{Y} = \mathbf{1}\{f(X, a) > \lambda_a\} + \mathbf{1}\{f(X, a) = \lambda_a, \ U > \rho_a\},$$

where $\rho_a \in [0,1]$ is determined by the selected rank among observations whose score is tied at the threshold. When there is no tie at $\lambda_a$, the second term is zero almost surely and the classifier reduces to the usual deterministic threshold rule.

This convention restores the uniform transform needed for the Beta order-statistic argument. Let $F$ be the cumulative distribution function of $Score$, and let $F(s^-) = \lim_{t \uparrow s} F(t)$. Define the randomized probability integral transform $Z = F(Score^-) + U\{F(Score) - F(Score^-)\}$. For an arbitrary distribution of $Score$, including discrete or mixed distributions, we have $Z \sim \mathrm{Unif}(0,1)$. Therefore, if $(Score_j, U_j)_{j=1}^n$ are i.i.d. and ordered lexicographically, then the transformed value of the $k$-th augmented order statistic satisfies $Z_{(k)} \sim \mathrm{Beta}(k, n+1-k)$. Consequently, the Beta-distributed order-statistic quantities used in the fairness proof continue to apply to the augmented ranks.

In the federated setting, this randomized tie-breaking rule is applied separately within each client and each subgroup $(Y = y, A = a)$. For client $i$, let $F_i^{y,a}$ denote the conditional CDF of the score $f(X, a)$ given $Y = y, A = a$. The resulting finite-sample fairness guarantee should therefore be interpreted as holding jointly over the randomness of the calibration sample and the independent tie-breaking variables. The continuity assumption is not needed for the fairness certification once this randomized convention is adopted.

