# OpenReview forum: "Distribution-Free Fair Federated Learning with Small Samples"
_SLADS/Section_C — Decision pending for SLADS_Section_C_

### Review · Reviewer_dTzL · 2026-06-14

**Summary Of Contributions:**

This paper studies fair federated learning in small-sample and heterogeneous-client settings. The authors propose FedFaiREE, a post-processing method that takes a pre-trained score function and selects group-specific thresholds to control fairness. The main focus is Equality of Opportunity, with extensions to Equalized Odds, label shift, multiple sensitive groups, and other fairness notions.

The main technical idea is to convert the fairness constraint into a rank-based problem. Each client computes local score ranks within subgroups defined by the label and sensitive attribute. The server then uses approximate rank information, together with order-statistic arguments, to construct a set of candidate classifiers satisfying the fairness constraint with finite-sample probability control. The use of Beta distributions for order statistics is the key statistical ingredient of the paper.

The paper also provides an accuracy guarantee relative to a fair Bayes-optimal classifier under additional assumptions on the score function and quantile stability. Experiments on several benchmark datasets show that FedFaiREE can reduce fairness violations with relatively small loss in accuracy.

Overall, the paper addresses an important problem and proposes a promising statistical approach. However, several theoretical and implementation details need to be clarified before the finite-sample fairness guarantee can be fully trusted.

**Audience:**

Yes

**Broader Impact Concerns:**

The paper addresses an important and socially relevant problem: improving fairness in federated learning without centralizing raw data. I do not see major negative broader-impact concerns. The proposed framework has potentially positive implications for privacy-sensitive applications.

**Claims And Evidence:**

Yes

**Requested Changes:**

1. Clarify the meaning of “distribution-free.” The paper should explain that the proposed guarantee is nonparametric and finite-sample, but it still depends on sampling assumptions, the mixture form of the test distribution, continuity of score distributions, and stability conditions used in the accuracy analysis.

2. Clarify the role of calibration data. The algorithm appears to use local training data to construct the rank-based calibration set. If the same data are used for both training the score function and calibrating the thresholds, the authors should justify the validity of the order-statistic argument. Otherwise, they should use sample splitting or cross-fitting.

3. Check the approximate rank bounds in Proposition 3.4. In the definition of the upper and lower approximate local ranks, the truncation appears to be in the wrong direction.  An upper rank bound should be capped above by $n_i^{1,a}+1$, while a lower rank bound should be bounded below by $0$. Thus, the truncations seem to require $\min$ for the upper bound and $\max$ for the lower bound. The authors should check these definitions in Proposition 3.4 and the related extensions, and clarify which version was used in the experiments.


4. Check and correct the definition of $L$ in the Equalized Odds algorithm. Based on Proposition 3.6, the violation probability upper bound should be the sum of the four nonnegative tail-probability terms. However, Algorithm 2 appears to define $L$ with minus signs. The authors should correct this definition and clarify whether the experimental code used the correct expression.

5. Clarify and correct the proof of Proposition 3.5. The current proof contains apparent conditioning and indexing errors in the decomposition of false-negative and false-positive probabilities. The authors should carefully check all four misclassification terms and verify that the final error bound is derived from the correct conditional distributions.

6. Explain how the method handles very small or empty subgroups. The authors should report subgroup sample sizes and specify what happens when some $n_i^{y,a}$ are zero or very small.

7. Discuss ties and discrete score distributions. The Beta order-statistic argument assumes continuous score distributions, but practical score functions may produce ties. The authors should specify the tie-breaking rule or provide a conservative extension.

8. Improve notation and proof presentation. Some formulas in the Equalized Odds extension and the appendix are difficult to follow. The authors may consider adding a notation table and checking the consistency of the notation and indexing to improve readability.

**Strengths And Weaknesses:**

Strengths:
The paper studies an important and timely problem. Fairness in federated learning is highly relevant in applications where data cannot be centralized and where local sample sizes may be small.
The proposed method is flexible. Since FedFaiREE is a post-processing procedure, it can be applied on top of different federated learning algorithms, such as FedAvg, FedFB, and FairFed.
The rank-based construction is statistically appealing. The use of order statistics gives the method a clear finite-sample interpretation. This is a useful contribution compared with purely penalty-based fairness methods.
The paper also takes communication constraints into account by using approximate quantile sketches rather than transmitting all local scores.

Weaknesses:
My main concern is the possible reuse of training data for fairness calibration. The algorithm appears to use the same local data to train the score function and to construct the rank-based calibration set. If so, the order-statistic argument may not apply directly, because the score function is no longer fixed independently of the calibration data. The authors should either use an independent calibration set, apply cross-fitting, or provide a justification for data reuse.

There also appears to be a possible error in the approximate rank bounds. The definitions of the upper and lower rank bounds seem to use max/min truncations in the wrong direction. This should be carefully checked, since this part is central to the fairness guarantee.

A more specific issue arises in the Equalized Odds extension. According to Proposition 3.6, the probability that the EO constraint holds is lower bounded by one minus four nonnegative tail-probability terms. Therefore, the corresponding upper bound on the violation probability should be the sum of these four terms. However, Algorithm 2 appears to define the candidate-set function $L$ with minus signs. Since $L$ is used to construct the feasible candidate set, this sign error may directly affect the validity of the algorithm and possibly the reported empirical results. The authors should correct the definition of $L$ and clarify whether the implementation used the correct expression.

I also found apparent conditioning errors in the proof of Proposition 3.5 in Appendix A.2. For example, when deriving $P_i(Y=1,\hat Y=0,A=0)$, the proof writes a term involving $P_i(\hat Y=1\mid Y=0,A=0)$ and an incomplete expression $P_i(Y=,A=0)$. This is not consistent with the target event. The correct decomposition should be
$$
P_i(Y=1,\hat Y=0,A=0) =
P_i(\hat Y=0\mid Y=1,A=0)P_i(Y=1,A=0).
$$
Similarly, the false-positive term $P_i(Y=0,\hat Y=1,A=0)$ should be conditioned on $Y=0,A=0$, but the proof appears to use $Y=1,A=0$ in several places. These may be typographical errors, but they occur in the proof of a key misclassification-error bound used later in the accuracy guarantee.

---

> ### Author Response · Authors · 2026-06-30
>
> **Response to Requested Changes 1**
>
> We thank the reviewer for pointing this out. In the revision, we clarify that “distribution-free” is used to denote a property holds with no distributional assumptions.: the fairness certificate does not implose any distributional assumptions for the client-specific conditional score distributions. We have added a clarification paragraph in Section 3.1.
>
>
> **Response to Requested Changes 2**
>
> Thank you for raising this important point. We agree that the wording in the algorithm was ambiguous. In the previous version, the input was described as ``Train dataset'' because FedFaiREE is a post-processing procedure applied after the score function $f$ has already been trained. However, this terminology could incorrectly suggest that the same data are used both to train $f$ and to calibrate the group-specific thresholds.
>
> To avoid this ambiguity, we have revised the notation throughout the paper and now refer to the data used by FedFaiREE as the calibration dataset. More specifically, we have updated Algorithms 1-4 and the related text to replace Train dataset' with Calibration dataset and to explicitly state that the calibration data are used only for computing subgroup scores, local ranks, and quantile sketches for threshold selection.
>
>
> **Response to Requested Changes 3**
>
> We thank the reviewer for carefully pointing this out. This is indeed a typo. In the revised paper, we have corrected these definitions, as well as the related extensions. We also clarify that the implementation used in the experiments followed the correct truncation convention. Therefore, the experimental results reported in the paper are unaffected by this correction.
>
>
> **Response to Requested Changes 4**
>
> We thank the reviewer for carefully checking the Equalized Odds algorithm. The minus signs in the definition of the violation probability bound in Algorithm 2 were a typo. As stated in Proposition 3.6, this quantity should be defined as the sum of the four nonnegative tail-probability terms. We have corrected Algorithm 2 in the revised paper accordingly. We also clarify that the experimental code used the correct summed expression, so this typo does not affect any of the reported experimental results.
>
> **Response to Requested Changes 5**
>
> We thank the reviewer for carefully examining the proof of Proposition 3.5. We agree that several intermediate lines in the previous proof contained typographical errors in the conditioning and indexing of the false-negative and false-positive probability decompositions. We have carefully checked all four misclassification terms and corrected these presentation errors in the revised paper. The corrected proof now uses the appropriate conditional distributions throughout. We emphasize that these errors were only in the intermediate derivation as written, and the final error bound and the statement of Proposition 3.5 remain unchanged and correct.
>
> **Response to Requested Changes 6**
>
> We thank the reviewer for raising this important practical issue. In the revision, we explicitly handle this case as follows. If $n_i^{y,a}=0$ for a client-level subgroup, we do not discard the client or treat the empirical subgroup frequency as evidence that the population subgroup mass is zero. Instead, the client’s subgroup CDF contribution is conservatively bounded by the full interval $[0,1]$, equivalently using the endpoint conventions
> $Q(0,1)=0, Q(1,0)=1$. This makes the certificate more conservative but preserves validity. For very small non-empty subgroups, the same Beta order-statistic construction is used; the resulting uncertainty are naturally larger when the subgroup sample size is small.
> We have also clarified that, if the global conditioning subgroup has zero mass, the corresponding conditional fairness metric is undefined rather than automatically satisfied. In such cases, the method reports the fairness metric as not applicable. Furthermore, if no threshold pair satisfies the certified constraint, FedFaiREE reports “no certified classifier found.” Any fallback classifier used for prediction is clearly marked as uncertified and is not used to claim the finite-sample fairness guarantee.

---

> ### Author Response · Authors · 2026-06-30
>
> **Response to Requested Changes 7**
>
> Thank you for pointing this out. We agree that the original manuscript did not sufficiently address ties and discrete score distributions. The proof used continuity through the probability integral transform, while practical score functions and quantile sketches may indeed produce tied scores. In the revision, we introduce an explicit randomized tie-breaking rule. Each score is augmented with an independent $U\sim Unif(0,1)$, and scores are ordered lexicographically by $(f(X,A),U)$. Equivalently, the final classifier randomizes only among samples whose score equals the selected threshold. Under this convention, the randomized probability integral transform $ F(Score^-)+U\{F(Score)-F(Score^-)\} $ is uniform for arbitrary score distributions, including discrete or mixed distributions. Hence, the Beta order-statistic argument remains valid without assuming that the original score distribution is continuous.
>
> We have added Appendix E to provide a detailed discussion of this randomized tie-breaking construction and to clarify that the finite-sample fairness guarantee is taken over both the calibration-sample randomness and the auxiliary tie-breaking randomness.
>
> **Response to Requested Changes 8**
>
> Thank you for the suggestion. In the revision, we have added a notation table in Appendix A.
> We also checked the consistency of the notation and indexing In addition, we added brief clarifications to make the relationship between global ranks, local ranks, and approximate rank bounds easier to follow. We hope these revisions improve the readability of the theoretical presentation.

---

### Review · Reviewer_7D5N · 2026-06-15

**Summary Of Contributions:**

The paper studies fairness in federated learning (FL) through a post-processing framework. The proposed method, FedFaiREE, extends FaiREE to decentralized settings and is designed to handle heterogeneous client distributions, making it more suitable for FL applications. The paper focuses on two fairness notions, Equality of Opportunity (EOO) and Equalized Odds (EO). The proposed algorithm seeks to maximize prediction accuracy while ensuring that the fairness violation probability is controlled below a prescribed level. The authors establish theoretical guarantees for both fairness and accuracy. Experimental results on semi-synthetic and real-world datasets demonstrate that FedFaiREE can effectively control fairness while maintaining competitive predictive accuracy with little performance loss.

**Audience:**

Yes

**Broader Impact Concerns:**

No broader impact concerns.

**Claims And Evidence:**

Yes

**Requested Changes:**

The main suggested improvements are described in the **Major** comments. Among them, I consider the first major comment to be the most important. Even a lightweight experiment on a single dataset with varying heterogeneity levels would be informative. If additional experiments are not feasible at this stage, I would still appreciate a more detailed discussion of the expected impact of heterogeneity on the proposed method. The authors may choose to address the **Minor** comments if time allows. However, these issues are **not critical** to my recommendation. The paper would still be acceptable in my view even if these points are only partially addressed.

**Strengths And Weaknesses:**

## Strengths

Strengths:

- The paper addresses an important problem in federated learning. Fairness in decentralized settings with heterogeneous data remains challenging and practically relevant.
- The proposed method provides theoretical guarantees for both fairness control and accuracy.
- The framework is general and can be extended to other fairness metrics, as demonstrated in the supplementary material.
- The experimental results show that the proposed method can substantially improve fairness while maintaining similar predictive accuracy.




## Weaknesses

### Major

- The authors claim that FedFaiREE is designed to address client heterogeneity. It would be helpful to provide experiments under different levels of heterogeneity and compare the performance of FedFaiREE with the baselines. Such experiments would provide more direct evidence for the claimed advantage under heterogeneous settings.
- More intuition would be helpful for Definitions 2.2 and 2.3, as these fairness metrics are the primary targets of the proposed method and are also used throughout the experimental evaluation. Simple examples based on the real datasets could improve readability.
- Similarly, Proposition 3.2 is one of the key results in the paper and serves as the foundation for the later fairness guarantees. Providing more intuition on the motivation and interpretation of this proposition would help readers understand the main idea of the method.
- Additional discussion of Figure 3 in the supplementary material would be helpful. Moreover, if the experiments were repeated multiple times, reporting standard deviations or error bars would provide a better understanding of the stability of the results.
- The conclusion from Table 11 in the supplementary material is not clearly discussed. From the results, it appears that the quality of the predictor $f$ mainly affects predictive accuracy, while the fairness control performance of FedFaiREE remains relatively stable. If this interpretation is correct, it would be valuable to explicitly discuss this finding, as it suggests that the proposed method remains applicable across a wide range of predictor qualities.

### Minor

- In Table 1, the proposed method does not appear to outperform FaiREE on the Compas dataset, unlike the results on the Adult dataset. Although I understand that FaiREE is not designed for decentralized settings, I am still curious why the relative advantage of FedFaiREE is less evident on Compas. Some discussion would be helpful.
- In Tables 2--4, FaiREE is not included in the comparisons. Since FaiREE is the most closely related method, some explanation of this choice would be helpful.
- If the running time and communication cost were recorded during the experiments, it would be helpful to report them. However, if these quantities were not recorded, I do not think additional experiments are necessary, and this comment does not affect my overall recommendation.
- In Section 3.1, Paragraph 2, Line 3, should the notation be $t_{i,j}^{y,a}=f(x_{i,j}^{y,a},a)$?
- In Section C of the supplementary material, a short roadmap describing the purpose of each subsection would improve readability.
- Figure 1 is presented in the Introduction before readers are familiar with the definition of DEOO. It may be helpful to revisit this figure in Section 5 when the corresponding experiments are discussed, so that readers can better understand its interpretation and significance.

---

> ### Author Response · Authors · 2026-06-30
>
> We sincerely thank the reviewer for the constructive and detailed comments. We also appreciate the reviewer’s acknowledgment of the strengths of our work.
>
> **Response to Major Weakness 1**
>
> Thank you for this important suggestion. We agree that experiments explicitly varying the level of client heterogeneity would strengthen the empirical evidence for our claimed advantage under heterogeneous federated settings.
>
> We have therefore conducted a new controlled experiment on the Adult dataset. Specifically, we vary the Dirichlet parameter $\omega$ over
> $$
> \omega \in \\{\mathrm{IID},10,1,0.3,0.1\\},
> $$
> where the IID is treated as a homogeneous baseline and smaller finite values of $\omega$ induce stronger client heterogeneity. The setting used in the original manuscript corresponds to \(\omega=1\), while the newly added settings allow us to evaluate FedFaiREE under both milder and more severe heterogeneity.
>
> Because the nominal Dirichlet parameter does not fully describe the realized partition, we additionally report
>
> $$
> H_A=\sum_i\frac{n_i}{n}
> \left|\widehat P_i(A=1)-\widehat P(A=1)\right|,
> $$
>
> which directly measures the weighted deviation between each client’s sensitive-group proportion and the global proportion.
>
> For each heterogeneity level, we use 100 clients and conduct 20 independent repetitions. For every seed and partition, the raw backbone, centralized FaiREE, and FedFaiREE use exactly the same trained scores and test data. We evaluate FedAvg, FedFB, and FairFed as backbone models. The fairness tolerance and confidence level are set to $\alpha=0.10$ and $1-\beta=0.95$, respectively.
>
> For a compact summary, the following table reports the worst result across the three backbones at each heterogeneity level. Full per-backbone results, including means and standard deviations, will be provided in the appendix.
>
> | Dirichlet $\omega$ | Realized $H_A$ | Worst $\lvert \mathrm{DEOO}\rvert_{95}$, FaiREE | Worst $\lvert \mathrm{DEOO}\rvert_{95}$, FedFaiREE | Worst violation rate, FaiREE | Worst violation rate, FedFaiREE | Max ACC diff. |
> |---:|---:|---:|---:|---:|---:|---:|
> | IID | 0.001 | 0.120 | 0.080 | 15% | 0% | 0.0008 |
> | 10 | 0.077 | 0.116 | 0.081 | 10% | 0% | 0.0007 |
> | 1 | 0.218 | 0.121 | 0.074 | 20% | 0% | 0.0004 |
> | 0.3 | 0.304 | 0.123 | 0.074 | 55% | 0% | 0.0008 |
>
> Here, the violation rate is the fraction of repetitions satisfying $|DEOO|>\alpha$ and Max ACC diff. denotes the maximum absolute difference in average accuracy between FaiREE and FedFaiREE across the three backbone models under each heterogeneity level. FedFaiREE keeps the empirical 95th percentile below the prescribed tolerance at every tested heterogeneity level, and its violation rate is always less than 5%, consistent with the target confidence level. In contrast, centralized FaiREE has at least one backbone whose 95th percentile exceeds the fairness tolerance at every level, with a violation rate as high as 55%.
>
> At the same time, the largest accuracy difference between FaiREE and FedFaiREE is below $0.001$. These results show that FedFaiREE achieves substantially more stable fairness control while maintaining essentially the same predictive performance as FaiREE.
>
>
> **Response to Major Weakness 2**
>
> We thank the reviewer for this helpful comment. We have revised Section 2, with the added text highlighted in blue, to provide more intuition for Definitions 2.2 and 2.3. In particular, we now explain why Equality of Opportunity and Equalized Odds are useful fairness metrics, and add an Adult income-prediction example to illustrate their practical meaning.
>
> **Response to Major Weakness 3**
>
> We thank the reviewer for this helpful suggestion. We agree that more intuition would improve readability. We have added a high-level explanation immediately after Proposition 3.2 to clarify the intuition behind the proposition and explain how it uses threshold ranks, local client information, and client weights to bound the probability of violating the DEOO constraint. This revision is highlighted in blue in the manuscript.
>
> **Response to Major Weakness 4**
>
> Thank you for this helpful suggestion. We have added an additional paragraph in the supplementary material to discuss and interpret Figures 3 and 4. The added discussion explains how the accuracy and $|DEOO|_{95}$ of FedFaiREE change with respect to $\alpha$ and $\beta$ on the Adult and Compas datasets. Due to space limitations, we do not report standard deviations in the main text. However, we report the corresponding standard deviations in Tables 7, 8, and 9 in the supplementary material.
>
> **Response to Major Weakness 5**
>
> Thank you for this insightful comment. Your interpretation is exactly correct. Table 11 shows that the quality of the predictor mainly affects predictive accuracy, while the fairness control performance of FedFaiREE remains relatively stable. To make this point clear, we have added a discussion immediately after Table 11 in the supplementary material, as highlighted in blue.

---

> ### Author Response · Authors · 2026-06-30
>
> **Response to Minor Weakness 1**
>
> We thank the reviewer for this helpful comment. The relative advantage of FedFaiREE is less evident on Compas mainly because both FaiREE and FedFaiREE are post-processing methods that adjust group-specific thresholds. Compared with Adult, Compas has a smaller sample size, and we also partition it into fewer clients. In this setting, the thresholds selected by FaiREE and FedFaiREE are nearly identical, which leads to the same performance in Table 1. We have added a brief discussion of this point in the revised manuscript.
>
> **Response to Minor Weakness 2**
>
> We thank the reviewer for this helpful comment. We agree that FaiREE is the most closely related method, and we should have explained this choice more clearly. FaiREE is a centralized post-processing method that requires pooled access to calibration scores, labels, and sensitive attributes. We have compared with FaiREE in Table 1. In Tables 2-4, we focus more on the federated setting, where the main comparison is with existing fair federated learning methods, such as FedFB and FairFed. The goal is to show that FedFaiREE provides stronger high-probability fairness control and, as a post-processing method, can also be combined with these federated baselines.
>
>
> **Response to Minor Weakness 3**
>
> We thank the reviewer for this helpful suggestion. We apologize that the running time and communication cost were not systematically recorded during the experiments. We will consider including these measurements in a future revision.
>
>
> **Response to Minor Weakness 4**
>
> We thank the reviewer for carefully pointing this out. This is indeed a typo, and we have corrected the notation in the revised manuscript.
>
>
> **Response to Minor Weakness 5**
>
> Thank you for the helpful suggestion. We have added a short roadmap at the beginning of Appendix C to describe the purpose of each subsection, with the new text highlighted in blue in the revised manuscript.
>
> **Response to Minor Weakness 6**
>
> Thank you for the helpful suggestion. We have revisited Figure 1 in Section 5 when discussing the corresponding Adult experiments and added a blue-highlighted explanation of its interpretation and significance.

---

> > ### Comment · Reviewer_7D5N · 2026-07-04
> >
> > Thank you for your reply. The authors have adequately addressed all my concerns.

---

### Decision · Action_Editor_RFZE · 2026-07-04

**Recommendation:** Accept as is

**Audience:**

The paper addresses the problem that is highly important and relevant to the SLADS Section C audience.

**Claims And Evidence:**

The paper is nicely put together with clear demonstrations of the new method.